# Trends in social vulnerability to storm surges in Shenzhen, China

Huaming Yu[1,2,3], Yuhang Shen[1], Ryan M. Kelly[4], Haiqing Yu[5], Xin Qi[6], Songlin Li[1]

[1]College of Oceanic and Atmospheric Sciences, Ocean University of China, Qingdao, 266100, China;
[2]Sanya Oceanographic Institute, Ocean University of China, Sanya, 572024, China;
[3]Key Laboratory of Physical Oceanography, Ministry of Education, Qingdao, 266003, China;
[4]Rykell Scientific Editorial, Los Angeles, CA, USA;
[5]College of Fisheries, Ocean University of China, Qingdao, 266003, China;
[6]Department of Organic Food Quality and Food Culture, Faculty of Organic Agricultural Sciences, University of Kassel, Nordbahnhofstrasse 1A, 37213, Witzenhausen, Germany;

*Correspondence to*: Haiqing Yu (yuhaiqing@ouc.edu.cn)
RMK, https://orcid.org/0000-0003-2322-2848
HY[4], https://orcid.org/0000-0002-0529-1172
XQ, https://orcid.org/0000-0002-2354-2828

**Abstract.** An evaluation of social vulnerability to storm surges is important for any coastal city to provide marine disaster preparedness and mitigation procedures and to formulate post-disaster emergency plans for coastal communities. This study establishes an integrated evaluation system of social vulnerability by blending a variety of single evaluation methods, which are subsequently combined by weighting in order to calculate a common social vulnerability index. Shenzhen has a current reputation of having the most considerable economic development potential and a representative city in China. The city is chosen for evaluation of its social vulnerability to storm surges via a historical social and economic statistical dataset spanning the period 1986–2016. Exposure and sensitivity increased slowly with some fluctuation, leading to some alterations of the social vulnerability trend. Social vulnerability keeps almost constant during 1986–1991 and 1993–2004, while it decreased sharply afterwards to form a 'stair-type' declining curve over the past 31 years. Resilience is progressively increasing by virtue of a continuous increase of medical services supply, fixed asset investments and salary levels of employees. These determinants contribute to the overall downward trend of social vulnerability for Shenzhen.

**Keywords:** Social vulnerability; Storm surge; Indicator system; Shenzhen, China;

## 1 Introduction

Storm surge refers to an abnormal volumetric rise of sea water layered above the astronomical tide due to severe meteorological conditions experienced during transition of low-pressure weather systems. Tropical and extratropical cyclones rank near the pinnacle among marine natural hazards in terms of human casualties and expensive infrastructure losses. As a naturally occurring phenomena, storm surge is a major contributor to coastal disasters and has significant

potential to disrupt communities, impair transportation systems, impact prosperous economic zones and reach record-
achieving damage levels. Most of the world's major coastal disasters caused by tropical cyclone activity are produced by resulting storm surge, such as Hurricane Sandy (2012) (Forbes et al., 2014; Rosenzweig and Solecki, 2014), Typhoon Haiyan (2013) (Lagmay et al., 2015; Needham et al., 2015; Yi et al., 2015), Cyclone Nargis (1972) (Fritz et al., 2009), Hurricane Irma (2017) (Xian et al., 2018), the Bhola Cyclone (1970) (Frank and Husain, 1971), and Hurricane Katrina (2005) (Fritz et al., 2007; Irish et al., 2008). To curb the escalating losses and casualties resulting from storm surge incidents and achieve
sustainable development, it is urgent for governments and local authorities managing coastal areas to carry out disaster risk prevention and reduction activities.

Storm surges typically range from tens of kilometers to thousands of kilometers, with time scales or cycles of about 1 to 100 hours. Storm surges can be divided into (i) typhoon storm surges and (ii) temperate storm surges. Both types of storm surges have an impact on China's coastal areas. In spring and autumn, the coastal area of the Bohai Sea is very susceptible to
the development of temperate storm surges. In summer, the southeast coast of China is frequently hit by typhoons and typhoon induced storm surges often occur. Therefore, storm surge disasters is a very serious matter to China, which is the country with the most frequent occurrences and suffers the most severe losses, among the coastal countries in the northwest Pacific Ocean (Zhao et al., 2007). Based on China's Marine Disaster Bulletin (1989−2008), Xie and Zhang (2010) pointed out that China's storm surge disasters are mainly concentrated in June to October each year, accounting for 88.19% of the
total economic losses from storm surge disasters. The spatial distribution of storm surge disasters shows that Guangdong, Zhejiang, Fujian and Hainan are the most affected provinces. From 1989 to 2008, the direct economic loss caused by storm surge disasters for these four provinces is 71.472 billion yuan, 58.584 billion yuan, 44.867 billion yuan and 33.09 billion yuan, respectively, accounting for 29.2%, 24%, 18.4% and 13.5% of the total economic loss caused by storm surges. Moreover, the annual maximum value of storm surge intensity tends to increase, and the direct economic loss caused by
storm surge disasters tends to fluctuate.

The occurrence of marine natural hazards depends not only on the hazards intensities but also on urban exposure and vulnerability (Dwyer et al., 2004; Peduzzi et al., 2009; Ellis, 2012; IPCC, 2012). Therefore, it is necessary to build detailed research involving human impacts and the positive effects when facing marine natural hazards (Cutter, 2003a). Risk assessment of tropical cyclone-induced storm surge provides the basis for risk mitigation and related decision making (Lin et
al., 2010). An effective coping with disaster risk requires a more rational distribution of efforts in areas such as disaster risk reduction and disaster management. Disaster reduction should be regarded as a new dimension of development rather than simply focused on post-disaster responses (Zheng et al., 2012). Whether a disaster is initiated by weather, climate or hydrological events, it can result in a tangible problem and depends largely on specific physical, geographical and social conditions (Sun et al., 2009; Yin et al., 2012). In this sense, vulnerability has become one of the central elements of
sustainability research (Turner et al., 2003a). Understanding, measuring, and reducing vulnerability has been one of the most important priorities in the transition to a more sustainable world (Birkmann, 2006). In comparison to other coastal disasters, there are few studies on the vulnerability to storm surge. Therefore, the ability to effectively evaluate the vulnerability to

storm surges is of great significance for reducing the consequences of this type of marine natural hazard.

At present, there is still no universal concept of vulnerability, though it is generally defined as the possibility, degree, or state of the system being damaged (Huang et al., 2012). It is widely understood that vulnerability is an inherent attribute of the system, and the state of the exposure factors in the risk of damage is the core characteristic of vulnerability (Cardona, 2004).

However, views about the components of vulnerability vary among disciplines and research areas (Dow and Downing, 1995; Cutter, 1996; Janssen et al., 2006). Based on the theory of sustainable development and from a disaster economics perspective, vulnerability of a system is identified by its ability to prevent and resist a disaster (Turner et al., 2003b). In the field of climate change, vulnerability refers to the degree to which a system is susceptible to, or unable to cope with, adverse effects of climate change, including climate variability and extremes (IPCC, 2012). Vulnerability is defined to be a function of the character, magnitude and rate of climate variation to which a system is exposed, its sensitivity, and its adaptive capacity (McCarthy et al., 2001; Adger, 2006).

Existing studies divide vulnerability into biophysical vulnerability, social vulnerability and an integrated vulnerability (Cutter, 2003a; Schmidtlein et al., 2008; Clare and Weninger, 2010). Biophysical vulnerability refers to a certain amount of (potential) loss of a system caused by a particular climatic event or hazard, which can be measured quantitatively by a series of indicators such as human death, production cost loss and ecosystem loss (Jones and Boer, 2005). While social vulnerability places more emphasis on its social connotation, focusing on the analysis from the perspective of the characteristics of a person or group in terms of their capacity to anticipate, cope with, resist, and recover from the impacts of a natural hazard is important (Dwyer et al., 2004; Wisner et al., 2004; Zhang and You, 2014). Social vulnerability is partially the product of social inequalities and is a function of the demographics of the population as well as more complex constructs, such as healthcare, social capital, and access to lifelines (Cutter and Emrich, 2006). The social and biophysical vulnerabilities interact to produce the overall place vulnerability (Cutter, 1996; Fuchs and Thaler, 2018). However, vulnerability is also strongly influenced by a society's dependence on critical infrastructure such as roads, utilities, airports, railways, and emergency response facilities (Aerts et al., 2014; Bevacqua et al., 2018). It is important to note that while reducing exposure and vulnerability may considerably reduce flood damage and entail lower investment costs, they do not prevent flood waters from entering any coastal city (Cutter et al., 2000).

Before the 1990s, considerable research attention was paid to components related to biophysical vulnerability, but relatively few studies were carried out on social vulnerability due to the fact that quantifying social vulnerability has higher complexity than biophysical vulnerability (Mileti, 1999). However, large losses of life and property resulting from the occurrence of more devastating disasters have brought up the attention on the role of social vulnerability in disaster impact (Zhou et al., 2014). People began to realize that simply understanding the characteristics of biophysical vulnerability is not enough to analyze the losses caused by disasters and the ability to quickly recover from the disasters (Schmidtlein et al., 2008). The evaluation of social vulnerability is thought to be an important step in disaster risk assessment (Wisner et al., 2004; Cutter and Finch, 2008). Hence, governments should analyze the social vulnerability of coastal cities in order to build

policies such as distributing relief funds and assist the region to improve its adaptation capacity against coastal disasters (Wei et al., 2004). Thus a considerable amount of research on social vulnerability has emerged as a component of studies on disaster reduction in the same period (Cutter, 2003a; Cutter and Emrich, 2006; Schmidtlein et al., 2008).

Analysis of social vulnerability to storm surges in Shenzhen, China during 1986–2016 is important due to four main reasons. First, there has been few assessments of social vulnerability to storm surges in which Shenzhen is considered. Therefore, by furnishing a comprehensive screening of social vulnerability to storm surges in Shenzhen, the research provides a buffer against disaster risk and allows the city's government to plan for a more sustainable future. Also, the statistical methods and concepts used in this research can be adapted to other coastal cities, which are exposed to similar or

other types of marine natural hazards. Secondly, since 1979, political reform and openness has led to rapid urbanization and socioeconomic development in Shenzhen. By choosing Shenzhen, we study a typical scenario of social vulnerability change as a result of the extensive progress of a highly developed city. Thirdly, so far, research involving vulnerability to disasters are mainly focused on discussing the spatial distribution of vulnerability, as well as comparing the differences between various geographic areas and development levels. Instead, herewith, a composite social vulnerability index (SVI) for Chinese

coastal cities was developed by integrating 17 indices from three aspects (i.e. exposure, sensitivity and adaptive capability) that shaped the social vulnerability of urban society to hazards and analyzed the differences of vulnerability of different areas (Su et al., 2015). Data envelopment analysis (DEA) was used for regional vulnerability evaluation in China to discover a significant negative correlation between the level of vulnerability and the economic level of the region (Huang et al., 2011). Five methods for combined evaluation were used by Liu and Liu (2017). Their results determined that among seven coastal

cities in Shandong province selected for evaluation, Yantai city and Binzhou city had the highest and lowest vulnerability, respectively. The socioeconomic vulnerability to typhoon-induced storm surges was assessed for municipal districts of Guangdong province using a fuzzy comprehensive evaluation method. It was determined that vulnerability presented a large spatial heterogeneity (Zhang et al., 2010). Research focused on the risk assessment of typhoon disasters in China's coastal areas by Niu et al. (2011) and research on the regional vulnerability of storm surge disasters by Yuan et al. (2016) led to

similar conclusions. However, the social vulnerability to storm surges contains both spatial and temporal dimensions. It is of significant value to observe the changes of social vulnerability over years for one disaster prone coastal city by identifying factors contributing to large impacts on social vulnerability, which in return, becomes beneficial for generating disaster prevention and mitigation policy.

     Thus, the purpose of our study is to quantitatively explore the trends of social vulnerability to storm surges in Shenzhen

from a macroscopic perspective. Based on the postulation put forward by Turner et al. (2003a), social vulnerability in our study is divided into three aspects: (i) exposure, (ii) sensitivity and (iii) resilience, so we can inspect the results from different perspectives.

## 2 Materials and methods

### 2.1 Study area and data sources

Shenzhen (22° 32' 34.3788" N, 114° 3' 46.7856" E) is a metropolitan city attributed to one of the highest gross domestic product (GDP) per capita in mainland China and its economic aggregate is equivalent to a medium-sized Chinese province (Zünd and Bettencourt, 2019). Since its establishment in 1979, in just 40 years, Shenzhen has gone through a tremendous advancement by virtue of political reform and a more open environment (Fig. 1c). Through the growth of GDP, it is found that Shenzhen's economic level is progressively advancing during the study period (Fig. 2).

However, due to its location at the coast of the Pearl River Delta (Fig. 1a,b) and its proximity to the northern part of the South China Sea (Fig. 1b,c), Shenzhen is facing many coastal disasters threatening its sustainable development, among which storm surge induced disasters are the most severe. According to the Shenzhen Marine Disaster Emergency Plan (2017), there have been 260 typhoons affecting the coastal areas of Shenzhen since 1949, with an average of 4.06 typhoons per year. Among them, 116 typhoons have seriously affected the Shenzhen coastal area with an average of 1.81 typhoons per year, especially typhoons landing in the coastal areas, causing the greatest impact within the city limits (Fig. 1c, crimson color coding). 13 typhoons have made landfall directly on Shenzhen's coastline and the strongest system was Typhoon "7908". Typhoon "7908" made landfall in the end of July 1979, which caused the storm surge level at Red Harbor to reach 1.12 m. On a broader perspective, the highest storm surge level ever recorded in China occurred with Typhoon "8007". Typhoon "8007" made landfall in July 1980 and generated a 5.94 m surge at Nandu Tide Gauge in Leizhou, China, a tide gauge notable for recording four out of the six highest water levels from coastal flooding situations (Liu and Wang, 1989; Ma, 2003; Zhang, 2009; Needham et al., 2015). The increased frequency of storm surges has caused ever growing economic and social losses in Shenzhen each year. Therefore, it is valuable to commence a risk assessment and develop an early warning system for Shenzhen in order to protect a particularly susceptible area from future storm surge impacts.

The data used to evaluate the social vulnerability of storm surges in Shenzhen is entirely available in Shenzhen Bureau of Statistics, Shenzhen Investigation Team of National Bureau of Statistics (2017), which is compiled and published on annual basis by the Shenzhen Statistical Bureau. Therefore, the instantaneity and reliability of this data are acceptable for research purposes. This yearbook comprehensively and systematically introduces the national economy and social development of Shenzhen, and the indicators reflect the achievements made by Shenzhen in all aspects of economy and society in 2016, as well as the statistical data of the city since its establishment. The statistical data consists of 19 parameters, listed as: (i) synthesis, (ii) national economic accounting, (iii) population and labor force, (iv) industry and energy, (v) construction industry, (vi) transport and post and telecommunications, (vii) agriculture, (viii) investment in fixed assets, (ix) real estate development, (x) commerce and prices, (xi) financial revenues and expenditures, (xii) financial insurance industry, (xiii) foreign economic trade and tourism, (xiv) labor wages, (xv) science and technology, (xvi) culture and education, (xvii) health, social security and social welfare, (xviii) urban construction and environmental protection, and (xix) people's livelihood. Due to the absence of long-term statistical data on some important indicators, this study is limited to a partial

statistical dataset spanning the period 1986–2016 in order to sustain the data integrity. Although including all factors to the indicator system for analysis would reach better agreement with the marine disaster community, this study can only provide certain factors due to data availability limits. For example, elderly people and people with disabilities are included in vulnerable groups (Yuan et al., 2016) which should be reflected in sensitivity, but there is no specific data captured about the

elderly population in Shenzhen's statistical yearbooks. In terms of study areas, the research limits coastal city choices based on several assumptions. Candidate cities should have (i) datasets with relatively complete, detailed statistics, (ii) well developed coastal industries such as agriculture, fishing, etc., (iii) a sharp, increasing population growth and matching economic development pattern, and (iv) suffer from frequent and severe storm surges. Additionally, non-candidate coastal cities are mature, populous cities with a long economic history and had slower development stages or primitive cities with a

slower economic growth rate and possess fewer established coastal industries. As a limit to the study, a fit method should be developed to determine which cities match specific criteria suitable for becoming appropriate candidates for this research.

**2.2 Research methods**

At present, the evaluation of social vulnerability is still in an exploratory stage and the theoretical frameworks used in various fields are dissimilar, such as the hazards of place (HOP) model (Cutter, 1996) and the vulnerability framework for

sustainability science (VFSS) model (Turner et al., 2003a), etc. Currently, the unified evaluation model has not been completely established (Zhou et al., 2014). Based on these frameworks, the existing social vulnerability assessment methods can be divided into three kinds: (i) based on an indicator system (Su et al., 2015), (ii) based on historical disaster loss (Sun et al., 2009), and (iii) based on a vulnerability curve. This paper adopts the first assessment method and is based on the SVI evaluation framework proposed by Cutter (1996), which is comprised of calculating the SVI to measure the vulnerability

level of a region by selecting the indicators related to the social vulnerability of that region (Cutter, 1996). The evaluation indicator system of disaster vulnerability is composed of two parts: (i) the indicator system and (ii) the indicator weight. The indicators reflect the characteristics of the evaluation objects and their internal relations while the indicator weight reflects the importance of the indicator to the final score and is an essential part of the construction of the evaluation system (Yang and Li, 2013). At present, the methods used to determine the weight of evaluation indicators can be divided into two

categories: (i) subjective weighting method and (ii) objective weighting method. The former is dominated by the expert grading method (Liu et al., 2002; Wang et al., 2003), while the latter encompasses several research methods, including the analytic hierarchy process (AHP) (Lu, 2008; Shi et al., 2008), principal component analysis (PCA) (Zhang and You, 2014), data fusion algorithms and the comprehensive analysis method (Liu and Liu, 2017). Among them, the comprehensive analysis method refers to the combination of two or more single evaluation methods to determine the indicator weight, which

enhances the objectivity and rationality of the evaluation results.

  Based on the above mentioned research, this study constructed a set of basic procedures for calculating the SVI of storm surges in Shenzhen (Fig. 3). Firstly, the construction of an optimized social vulnerability evaluation indicator system, based on the idea of rough set theory (Das et al., 2018), is completed. Second, the entropy method (Zhou and Yang, 2019), the

technique for order preference by similarity to an ideal solution (TOPSIS) method (Kuo, 2017) and the coefficient of
variation method (Zhou et al., 2004) are used to weigh the indicators and aggregate SVI separately. Then, the consistency of
different evaluation results is tested by using the compatibility test method, i.e., Kendall consistency test (Wen and Hu,
2002). When all the above evaluation methods pass the consistency test, the combination weighting method is used to
determine the weight of each evaluation method. Finally, the combined evaluation results are achieved, which have
significant advantages compared to those of all single methods due to weighted value of each evaluation method.

The analysis of the connotation and extension in the concept of vulnerability evaluation for a storm surge-bearing body is
based on vulnerability theory. Next, the evaluation indicators are preliminarily selected based on the perspective of exposure,
sensitivity and resilience and the indicator designing principles of science, system, dominance, comparability, quantifiability,
operability and dynamics. Finally, the evaluation indicators are screened and the optimal evaluation index system is
constructed by using the information extraction ability of rough set.

Rough set theory is a soft computing technique proposed by Z. Pawlak for handling vague, inconsistent and uncertain
data (Das et al., 2018). The main idea is to remove redundant or unimportant attributes according to specific rules on the
premise of keeping the classification ability of knowledge base unchanged (Wu and Tang, 2019). This method can undertake
in-depth analysis and reasoning of data, simplify the data, and obtain knowledge on the premise of preserving key
information, identify and evaluate the dependencies between the data, and finally, reveal the potential regularity from the
data (Pawlak, 1998; Pawlak and Skowron, 2007). Rough set is defined in terms of a pair of sets, namely lower
approximation and upper approximation of the original set. Indiscernibility relations and set approximations are the
fundamental concepts of the rough set theory (Pawlak, 1982; Swiniarski, 2001).

In order to enhance the reliability of the social vulnerability evaluation results, it is inadvisable to apply only one
evaluation method. Therefore, this paper will use the entropy, TOPSIS and coefficient of variation methods to weigh the
social vulnerability indicators and aggregate SVI, respectively. When the calculation results of all evaluation methods in use
pass the Kendall consistency test, their combined evaluation results based on the combination weighting method are
achieved. The results under a single evaluation framework (i.e., the combination weighting method) will be further
investigated.

### 2.2.1 Entropy method

In information theory, entropy is a measure of uncertainty. The greater the amount of information, the smaller the uncertainty
and the entropy. According to the characteristics of entropy, we can determine the randomness and disorder degree of an
event by calculating the entropy value, or the entropy value can be applied to judge the dispersion degree of an indicator. The
greater the dispersion degree of an indicator, the greater the influence of this indicator on the comprehensive evaluation
(Skotarczak et al., 2018). Therefore, the weight of each indicator can be calculated according to their variation degree, using

information entropy as a tool to provide the basis for a comprehensive evaluation of multiple indicators (Zhou and Yang, 2019).

**Procedure I**

- **Step 1:** Select $n$ years and $m$ indicators.

- **Step 2:** Calculate the proportion of the indicator $j$ in year $i$ ($r_{ij}$):

$$\overline{r_{ij}} = \frac{r_{ij}}{\sum\limits_{i=1}^{n} r_{ij}} \quad , \tag{1}$$

- **Step 3:** Calculate the information entropy (e) of the indicator $j$:

$$e_j = -\left(\ln n\right)^{-1} \sum_{i=1}^{n} \overline{r_{ij}} \ln \overline{r_{ij}} \, (0 \le e_j \le 1, j = 1,2,3,\cdots,m) \quad , \tag{2}$$

where, $0 \le e_j \le 1$ and $j = \left[1,2,3,\ldots,m\right]$ .

- **Step 4:** Calculate the utility value of the indicator $j$:

$$d_j = 1 - e_j \quad , \tag{3}$$

- **Step 5:** Calculate the weight of the indicator $j$:

$$u_j = \frac{d_j}{\sum\limits_{j=1}^{n} d_j} \quad , \tag{4}$$

- **Step 6:** Obtain the final evaluation value by weighted summation of each indicator.

**2.2.2 TOPSIS method**

The TOPSIS method, namely the solution distance method, was first proposed by C.L. Hwang and K. Yoon in 1981 (Kuo, 2017). TOPSIS is a common multi-indicator and multi-objective decision analysis method, which has been widely applied to the evaluation of multivariate analysis (Wu and Chen, 2019). Its core idea involves sorting the proximity of a limited number of evaluation objects to idealized targets by measuring the distance of the positive ideal solution and negative ideal solution,

and then realize the evaluation of each object relative merits (Lu et al., 2011).

The TOPSIS method is performed in six steps, which are: (i) construct the original data matrix, (ii) data standardization processing, (iii) determine the indicator weight using the entropy method, (iv) calculate the positive and negative ideal values, (v) calculate the distance from each evaluation indicator to the positive and negative ideal value, and (vi) calculate the relative proximity between the evaluation object and the optimal value (Zhang and You, 2014).

**2.2.3 Coefficient of variation method**

A comprehensive evaluation is carried out through multiple indicators. When the value of an indicator can clearly distinguish each sample, the indicator possesses resolved information about this evaluation. Therefore, in order to improve the discrimination validity of a comprehensive evaluation, the idea of the coefficient of variation method is to assign weights to all the evaluated objects according to the variation degree of the observed values of each indicator (Zhou et al., 2004).

Indicators with large variation of the observed values indicate that the schemes or indicators can be effectively divided, and a larger weight should be given, otherwise a smaller weight would be justified (Zhao et al., 2013). The variation information of indicators is measured by its variance, but the variance of indicators is not comparable due to the influence of the dimensions and order of magnitude of each indicator. Therefore, the comparable indicator variation coefficient should be selected and the weight of each indicator can be obtained by normalizing its coefficient of variation (Gupta and Gupta, 2016).


**Procedure II**

- **Step 1:** Suppose there are $n$ participating samples, each of which is described by $p$ indicators. Calculate the mean value $X_{avg}$ and variance $S_i^2$ of each indicator.

$$X_{avg} = \frac{1}{n \sum X_{ij}} \quad , \tag{5}$$

$$S_i^2 = \frac{1}{n-1} \sum \left( x_{ij} - X_{avg} \right)^2 \quad , \tag{6}$$

- **Step 2:** Calculate the coefficient of variation of each indicator.
$$V_i = S_i / X_{avg} \quad , \tag{7}$$

where, $i = \left[ 1,2,3,\ldots,p \right]$ .

- **Step 3:** Obtain the weight of each indicator by normalizing the coefficient of variation.

$$W_i = \frac{V_i}{\sum V_j} \quad , \tag{8}$$

where, $j = \left[ 1,2,3,\ldots,p \right]$ .

- **Step 4:** Obtain the final evaluation value by weighted summation of each indicator.

**2.2.4 Kendall consistency test**

Due to limitations of the methods in use, each single evaluation can lead to a different conclusion. Nevertheless, as long as the evaluation criteria are consistent, the result of grade classification is reasonable. The Kendall consistency test is a method to test whether the results of each single evaluation method are consistent (Wen and Hu, 2002).

$$W = \frac{\sum_{i=1}^{n}\left(R_i - \frac{m(n+1)}{2}\right)^2}{m^2 n(n^2-1)/12} \quad , \tag{9}$$

where, $W$ is the Kendall's coefficient of concordance, $m$ is the number of evaluation methods used, $n$ is the year participated in the evaluation, and $R_i$ is the rank sum of year $i$. The numerator in Eq. (9) is the sum of deviation squared between the total rank and the total rank of all samples, and $n(n^2-1)/12$ in the denominator is the sum of total deviation squared (total sum of squares) of all ranks.

The closer $W$ is to 1, the greater the difference between the rank groups, wherefore there is a significant difference in the
scores of the years involved in the evaluation and further indicates that the evaluation criteria of different methods are consistent. On the contrary, the closer $W$ is to 0, the more inconsistent these methods are in their evaluation criteria.

**2.2.5 Combination weighting method**

In a single evaluation system, the results may possess slight one-sidedness differences, which will affect the accuracy and
feasibility of the evaluation. By combining the evaluation results of multiple evaluation methods helps to safeguard the objectiveness of the evaluation results.

A weight combination strategy normalizes the weight of a single method vector by using dispersion maximization combined with the weighting method in Eq. (10) and provides combination weight coefficients of singular evaluation methods. The combination weight of each indicator is obtained by using the combination calculation formula:
$\omega_s = \theta_1^* \omega_{1s} + \theta_2^* \omega_{2s} + \ldots + \theta_n^* \omega_{ns}$ , where $\theta_n^*$ is the weight of a single evaluation method, $\omega_{js}$ is the weight value of indicator $s$ under method $j$ ( $j = [1,2,\ldots,n]$ ), and $\omega_s$ is the final weight. In the following formula (Eq. 10), $f_{ij}$ , $f_{tj}$ are evaluated values of objects $i$ and $t$ under each single evaluation method ($j$), and $\theta_j^*$ is the weight of a single evaluation method ( $j = [1,2,\ldots,n]$ ):

$$\theta_j^\star = \frac{\sum\limits_{i=1}^{m}\sum\limits_{t=1}^{m}\left|f_{ij}-f_{tj}\right|}{\sum\limits_{j=1}^{n}\sum\limits_{i=1}^{m}\sum\limits_{t=1}^{m}\left|f_{ij}-f_{tj}\right|} \quad , \tag{10}$$


### 2.3 Indicator system of social vulnerability evaluation

By analyzing the factors contributing to social vulnerability, a set of more than 100 evaluation indicators was obtained (Fischer et al., 2002; Wisner et al., 2004; Zhou et al., 2014; Yuan et al., 2016). The evaluation indicators were then simplified using rough set theory.

The research screens an algorithm without considering the effects of man-made physical barriers and coastal defense systems such as seawalls, revetments, floodgates and dams. The algorithm screens for classifying a disaster body of interest (i.e., Shenzhen, China) that impact the social economy of the study area and screens for determining key attributes that can affect the exposure of a disaster body. Then, the evaluation indicators are selected based on aspects of both population and industrial structure to reflect the degree of sensitivity of a disaster body. Evaluation indicators are selected from aspects such

as fiscal expenditures, resident income, and infrastructure construction to reflect the resilience of a disaster body's social and economic system. Table 1 shows a total of 16 evaluation indicators selected after repeated screening in which the Grade I indicators are identified with the three components of vulnerability and the Grade II indicators – with the branches of the Grade I indicators.

     The indicators of exposure reflect the damage of an inundation area, including its population and social economy. Among

them, the permanent resident population at the end of the year reflects the population exposure. The higher the population, the higher the number of people exposed to natural disasters, and the relative high level of vulnerability. Since the amount of regional GDP measures economic exposure, a relative high level of economic development corresponds to a more vulnerable area to storm surges due to the aggregation of public property (e.g., shopping centers, office buildings, etc.) built upon the area compared to underdeveloped locations. In flooded areas, crops are damaged, fishery resources are affected and the port

cannot operate normally. The total area of crops, fishery output value and port cargo throughput are indicators directly exposed to the impact of storm surges.

     Sensitivity indicators reflect the degree of sensitive of a disaster body of interest (i.e., Shenzhen, China). Primary industries include agriculture, forestry, fishery, animal husbandry and collection. The operation of these industries is sensitive to changes of the natural environment and the occurrence of storm surges will directly affect their output of these

industries. When storm surges occur, surface meteorological conditions are harsh and often accompanied by severe winds and precipitations, which causes the city traffic to become busy and prone to accidents. Representing vulnerable societal groups, students and women are more likely to be injured or even to suffer casualties outside (Yuan et al., 2016). Meanwhile, social workers generally work outdoors with relatively high risk of being injured and their awareness of disaster prevention

and reduction is relatively low due to limited knowledge of the general population, leading to increased sensitivity of storm surges within the entire region.

In contrast to exposure and sensitivity, resilience is a negative indicator meaning that relatively high resilience in a region is equivalent to a relative low vulnerability. The resilience indicators selected for this research can be divided into three groups, namely (i) fiscal expenditures, (ii) resident income and (iii) infrastructure construction. Fiscal expenditure levels mainly reflect the general public budget expenditures and urban fixed asset investments. The higher the public budget spending, the more resources are provided/spent for social management and infrastructure development. Urban fixed asset investments include many infrastructure projects such as railways, water conservancy, roads, airports, pipelines and power grids. The higher the urban fixed asset investment values, the more developed the regional infrastructure is for a particular region. Therefore, with an increase of fiscal expenditures, the infrastructure construction is more complete and the ability to prevent and resist disaster consequences, along with resilience after being damaged, is substantial. The level of residential income can be divided into (i) disposable income of urban residents per capita and (ii) the average annual salary of employees. With a relatively high income level of residents and relatively higher living standard, the disaster resilience of the area becomes stronger and the recovery capacity is faster after the disaster (Yuan et al., 2016). The level of public services mainly refers to the level of medical and health care, including the number of medical and health institutions and their equipment (e.g., beds, etc.) as well as the number of health employees. All of these values are positively correlated with the medical treatment level of the potential victims.

## 3 Results and discussion

### 3.1 Variation pattern of social vulnerability

Based on the constructed evaluation indicator system along with detailed and reliable statistical data and combined weighting results, the annual SVI of Shenzhen between 1986 and 2016 is obtained and the changing characteristics and influencing factors of social vulnerability will be discussed. According to the common idea of equal division in mathematical statistics, degrees of social vulnerability to storm surges discussed in this research are set to (i) high vulnerability, (ii) relatively high vulnerability, (iii) moderate vulnerability, (iv) relatively low vulnerability and (v) low vulnerability and the corresponding critical thresholds of SVI are 0.5715, 0.5237, 0.4759 and 0.4281, respectively (Yuan et al., 2016).

According to calculated results, three kinds of single evaluation methods share similar weight coefficients, and the weight coefficients of the entropy method is the highest (Table 2). These results closely reflect a similar overall trend except for slight differences in numerical values. The combination of all three weighted values can be considered as a valid reflection of regional social vulnerability and used within the actual social vulnerability analysis.

As shown in Fig. 4, the weighted SVI exhibits a well pronounced overall downward trend (–0.006 per year) with noticeable fluctuations. SVI shows a slight upward trend between 1986–1991 and 1996–2004 and shows a significant downward trend (–0.04 per year) for the remaining years as the rate of decline is greatest within 2014–2016. According to

classification criteria, social vulnerability to storm surges in Shenzhen during the entire study period can be divided into five stages: (i) high social vulnerability between 1986 to 1994 and 1999 to 2004, (ii) relatively high social vulnerability between 1995 to 1998 and 2005 to 2008, (iii) moderate social vulnerability between 2009 to 2013, (iv) relatively low social vulnerability in 2014 and (v) low vulnerability in 2015 and 2016. Thus, the high social vulnerability stretched over the longest period of time opposed to the low vulnerability, which was only observed during the last two years of the study period. It is apparent that, after 2008, social vulnerability has been completely removed from relatively high levels.

The interdecadal changes of social vulnerability are also significant. Since 1986, each decade represents a cycle which has a step–down trend, and the derivative of the third step is the largest. By evaluating and classifying social vulnerability quantitatively, it is discovered that social vulnerability has been decreasing consistently during the research period. The discovered trend relates to Shenzhen's enhanced ability to withstand losses and recover after substantial damage when confronted with storm surges. The reasons for this trend has to be analyzed by the standpoints of exposure, sensitivity and resilience.

## 3.2 Reasons for vulnerability changes

Fig. 5 depicts the corresponding indices of exposure, sensitivity and resilience. It is important to note that exposure and sensitivity belong to benefit indicators which means the larger the exposure index (EI) and sensitivity index (SI), the higher the exposure and sensitivity. While resilience possesses opposite attributes as a cost indicator, meaning the larger the resilience index (RI), the lower the resilience.

The results show that exposure, sensitivity and resilience are increasing over time, as the growth rate in turn is resilience > exposure > sensitivity, which reflects that Shenzhen's social and economic exposure, sensitivity of population, and industrial structures have increased inevitably, but simultaneously. Shenzhen's fiscal spending, residents' income levels, completion degree of medical conditions, and infrastructure exponentially improved.

According to the evaluation results, the continuous increase of resilience is the most significant feature, which is mirrored by the continuous decrease of RI (Fig. 5). Resilience is closely related to the level of regional social and economic development. The remarkable pace of Shenzhen has greatly promoted the city's development in just thirty years which leads to a continuous growth of all resilience indicators. Therefore, the growth of resilience in Shenzhen is overt.

EI remains almost constant during the period 1986–1991 and, after presenting a slight drop between 1992 and 1996, continues growing since 1996. Shenzhen transformed from a small fishing village to grids of high-rise buildings after the rapid urbanization that followed the reform and openness policy occurred in 1979. This has led to a continuous decreasing trend of the exposure indicator (i.e., total sown area of crops; Fig. 6). In 1992, Deng Xiaoping delivered a famous speech during his inspection tour of south China. Afterwards, Shenzhen entered the second stage of speeded economic growth, during which better protected buildings and factories were built on what used to be farmland, causing the proportion of agriculture to decrease sharply. Consequently, the total sown area of crops reduced by less than one half of the previous year.

However, the weight of the 'total sown area of crops' indicator was relatively large (Table 3), which directly led to a decrease of exposure of Shenzhen during the same period.

Although the growth rate of SI is the lowest, SI maintains an upward trend except for a small decline between 2001 and 2006 because the 'proportion of females' indicator did not always increase with time. Instead, the proportion of females indicator showed a significant decreasing trend until 2006, which than sharply increased in a 10-year period (Fig. 6). In the

405 entire research period, SI is smaller than EI (Fig. 5) because the total weight of sensitivity indicators is the smallest (Table 3).

In Table 3, the weight of the indicators by benefit and cost types is very similar, accounting for approximately 50% of the total weight. Collectively, RI is larger than the sum of EI and SI. The statistical data corresponding to the resilience indicators are generally larger than that of exposure and sensitivity after standardization. The indicator weight is positively correlated with the dispersion of data, while the correlation coefficient between the indicator value and SVI is a measure of

410 degree of influence of this indicator on the social vulnerability. The first three indicators with the largest correlation coefficient are determined to be the number of medical and health institutions, urban fixed asset investments and annual average annual salary of employees, respectively. After data standardization, the three indicators are compared with the SVI (Fig. 7), and it is discovered that their trend is highly consistent. Three indicators that contribute to the greatest impact on SVI are all resilience indicators, indicating that social vulnerability for a region is more affected by its resilience while its

exposure and sensitivity only act a secondary binding role under the same development level. Moreover, in terms of the social vulnerability evaluation indicator system, the number of medical and health institutions are the most important resilience indicators that greatly influence the regional vulnerability, which reflects the ability for the region to treat injured people after a significant storm surge. The number of medical and health institutions reduced sharply in 1996 as the vulnerability index reached a minimum, concurrently.

**3.3 Validation of SVI to storm surges**

Economic loss data due to storm surges in Shenzhen is unavailable and a broader scale dataset was used in the validation of SVI to storm surges. Storm surge economic loss data spanning from 1991–2015 for Guangdong province was obtained from China's National Marine Bulletin (Bulletin of China Marine Disaster, 2018). The sum of the average of the peak speed and

425 landfall speed of typhoons combined with the extreme sea surface heights affecting Guangdong province each year is designated as the intensity of the storm surge in Guangdong province each year. Intensity and loss was adjusted to a range of 0.4–0.7 through standardization in order to match the range of SVI (Fig. 8).

Through data fitting, the relationship among storm surge intensity in Guangdong province and storm surge induced social vulnerability in Shenzhen between 1991–2015 is obtained. The best fit equation reads:

$$\text{loss} = 0.01282 + 0.7023 * \text{intensity} + 0.1986 * \text{SVI} \tag{11}$$

It is reasonable that storm surge loss is directly proportional to SVI and storm surge intensity.

The accuracy and reliability of Eq. (11) is verified in Fig. 9, where the theoretical loss (blue line) is calculated by the fitting equation and the real loss (red line) are shown. The trends of the two lines are similar to the correlation coefficient (CC; 0.7) and root mean square error (RMSE; 26 billion yuan) but the real loss fluctuates more than the theoretical loss (Fig. 9). In general, the fitted results are satisfactory from a macroscopic perspective and the reliability of Eq. (11) is considered high. The fitted equation determines that loss is positively correlated with both SVI and intensity, which provides evidence of an important connection between SVI and storm surges.

**4 Conclusion**

This research evaluates social vulnerability to storm surges in Shenzhen, China, from a macroscale perspective using 31 years of economic statistical data and 25 years of loss data. In accordance to the characteristics of storm surges and the connotation of social vulnerability, the study establishes the indicator system for social vulnerability evaluation respectively from three aspects: (i) exposure, (ii) sensitivity and (iii) resilience, based on the idea of rough set. The final weighted SVI is validated to be rational and reliable by combining results from multiple evaluation methods, based on the idea of combination weighting, in order for the results to objectively reflect the connotative information of social vulnerability in the indicator system.

The evaluation results show that the social vulnerability to storm surges in Shenzhen from 1986 to 2016 depicts a steady downward trend, with relatively pronounced interannual and interdecadal variability. The trend experiences four stages, passing through high to low social vulnerability, among which the period of relative high social vulnerability is the longest in duration. When analyzing the reasons for social vulnerability changes from exposure, sensitivity and resilience perspective, it is revealed that the increase of the social economy exposure and demographic and industrial structures sensitivity are less important than the disaster resilience. Therefore, with a large increase in resilience, the social vulnerability to storm surges in Shenzhen continues to decrease while the capacity to withstand and response to disasters has significantly improved.

The three most relevant indicators of social vulnerability belong to (i) resilience, which are the number of medical and health institutions, (ii) urban fixed asset investments and the (iii) average annual salary of employees. The study concludes that the increase of residents' income, infrastructure enhancement and medical and health conditions improvement are of great value to reduce social vulnerability.

Reducing social vulnerability is as valuable as sustainable development, as society is advancing and the economy continues to grow. The situation becomes inevitable as assets are exposed to disasters and populations vulnerable to substantial damage due to marine natural hazards are going to increase based on the theory of social vulnerability. This would lead to an increase in regional exposure and sensitivity. However, the general fiscal spending on public security of high investments, the increase of the residents' income levels, the improvement of the infrastructure, and the improvement of medical and health conditions are positive results of social progress. The relatively higher these indicators reach, the relatively lower the possibility of damage to a region materializes, and the stronger the disaster flexibility. This indicates that

the establishment of disaster prevention and reduction mechanisms for storm surges should mainly start from improving resilience through reasonable arrangements of financial expenditures, improving the living standard of residents and improving the infrastructure for disaster prevention. It is relatively difficult to reduce exposure and sensitivity, but their growth rate can be controlled by reducing crop acreage in areas vulnerable to storm surges, managing fishery breeding areas and the number of harbors, and selecting rational sites for residential areas and schools. In addition, the government should energetically develop more science and technology avenues, improve the mechanisms of marine forecasting to carry out real-time monitoring of future storm surges, closely monitor the tidal level changes at coastal tide stations, and issue storm surge early warnings through radio, TV and Internet channels in a timely fashion. All departments should strengthen communication and cooperation, establish and improve the response mechanisms to coastal disasters, and improve the emergency planning of storm surge incidents. After a coastal disaster occurs, governmental departments should assess all aspects of the damage levels, and provide completeness in post-disaster repairs to infrastructure.

Assessment of social vulnerability to storm surges is an important basis for disaster risk prevention, preparedness and reduction, as well as to formulate marine policy for emergency planning operations. However, some indicators were not included in the final evaluation system due to the lack of statistical data, such as coastal breakwaters, flooding areas, insurance depth and housing values. To further increase the reliability of the social vulnerability evaluation results, additional methods (e.g., fuzzy cluster analysis, PCA, efficacy coefficient method, expert evaluation method, etc.) and a greater number of methods deployed should be included in the research. Additionally, it is obvious that the scale of the social vulnerability evaluation at the municipal level is not as detailed as smaller administrative units, such as districts, residential quarters and streets. As an extension to this research, further challenges are related to narrowing of the evaluation scale of social vulnerability and selection of more reasonable indicators according to the local conditions.

**Data availability.**

The authors thank the Shenzhen Statistical Bureau and the National Bureau of Statistics for use of the historical 31-year dataset hosted in their Shenzhen Statistical Yearbooks. Yearbooks are available from the following website: http://www.sz.gov.cn/cn/xxgk/zfxxgj/tjsj/tjnj/ in PDF format (e.g., 2018 publication, http://www.sz.gov.cn/cn/xxgk/zfxxgj/tjsj/tjnj/201812/P020181229639722485550.pdf). The authors thank the Ministry of Natural Resources of the People's Republic of China for use of the historical 25-year dataset hosted in the Bulletin of China Marine Disaster. Bulletins are available from the following website: http://www.mnr.gov.cn/sj/sjfw/hy/gbgg/zghyzhgb/ (e.g., 2017 bulletin, http://gc.mnr.gov.cn/201806/t20180619_1798021.html; 2010 bulletin, http://gc.mnr.gov.cn/201806/t20180619_1798014.html). Figure 1 was created with QGIS 3.4 LTR, Python scripting with relevant mapping libraries, GIMP image editor for subplot modification, and LibreOffice Impress for figure organization. Figures 2, 4, 5, 6 and 7 were generated strictly with Python scripts. Figure 8 and 9 were generated strictly with MATLAB scripts.

**Author contributions.**

HY$_1$ and YS originated the idea, developed the methodology, analyzed the data and wrote the paper. HY$_1$ and YS conducted the main literature review. RMK added literature review support and modified parts of the manuscript with citations. RMK
wrote and refined Python scripts to produce the reference maps (Fig. 1) and graphs (Fig. 2, 4–7), and used LibreOffice Impress to construct the procedures diagram (Fig. 3). RMK used open source software, QGIS 3.4 LTR, to translate and edit a series of spatial data into an applicable map projection and geographic form, before reading it into Python scripts. YS produced the MATLAB scripts to create Fig. 8 and 9. YS compiled all tables and was involved with the refinement of all figures. XQ, SL and HY$_4$ assisted in data inquiries and analysis throughout the research period. RMK polished the paper with
detailed, multi-iterative English editing and proofreading stages. HY$_1$, YS and RMK were involved with the final checks of the manuscript. Note: The subscript near the initials stands for the author's position in the list.

**Competing interests.**

The authors declare that they have no conflict of interest.

**Acknowledgements**

This work was supported by the National Key Research and Development Program of China (Grant Nos. 2016YFC1401800,
2016YFC1402000, 2016YFC1401400, 2018YFB1502800) and the National Natural Science Foundation of China (Grant No. 41930534). The authors appreciate the anonymous reviewers for their discussions, comments and suggestions.

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

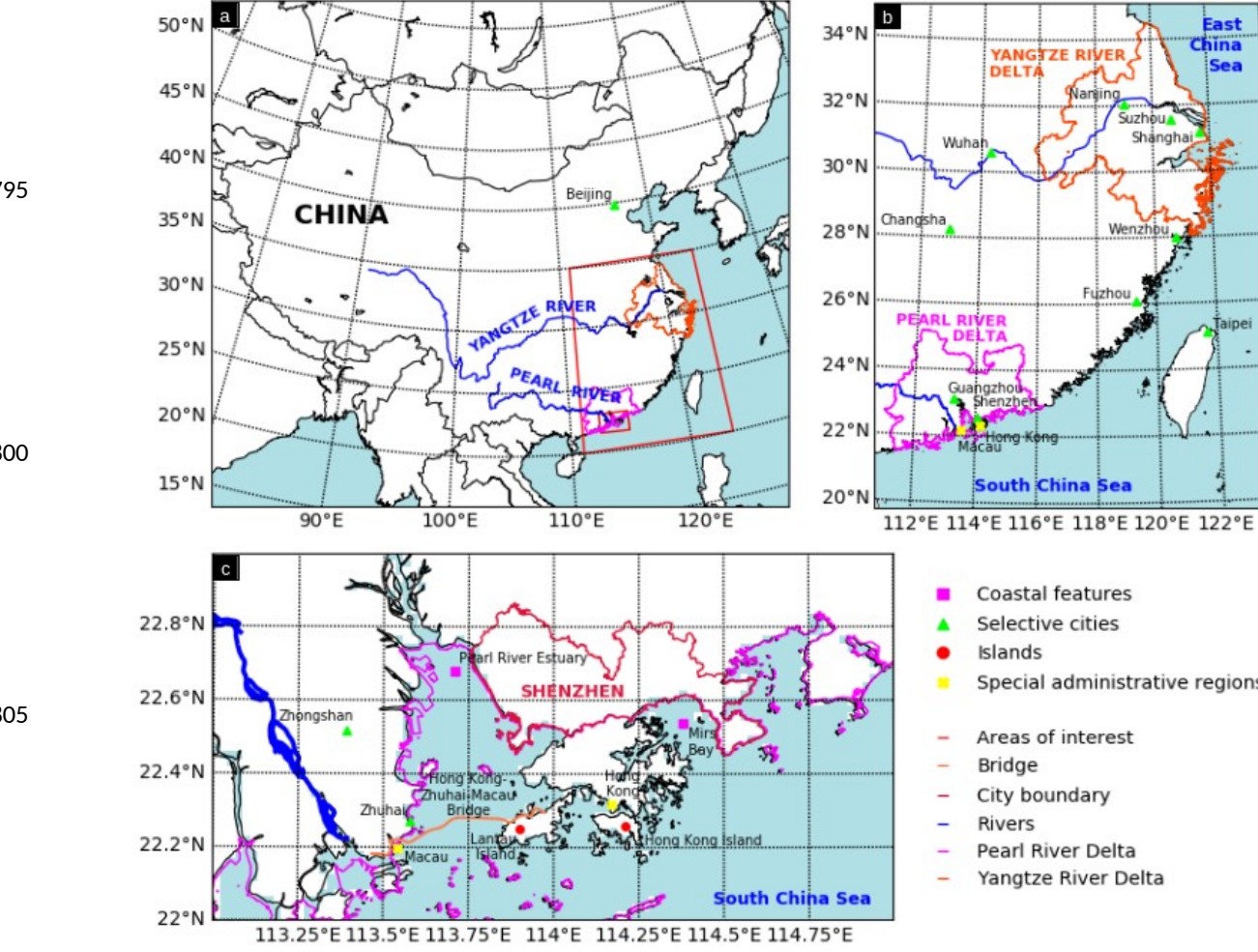

**Figure 1:** Mapped geographic features, shown at three scales: country wide (a), southeastern regional (b) and localized to the economic center of Shenzhen, China (c), are presented as a source of reference. The study area (Shenzhen, China) is labeled and outlined using crimson color in Fig. 1c. The maps apply the Lambert Conformal Conic (LCC) projection due to the country's middle latitude presence and predominantly east-west expanse. The LCC projection offers flexibility in adjustable standard parallels for plotting at different scales, where conformality is held true, angular distortion at any parallel (except for the poles) is essentially zero and meridians are right angles (Snyder, 1987). The LCC projection emphasizes the conceptual quality of secancy for conics and has been the conformal projection of choice for mid-latitudes (Pearson II, 1990).


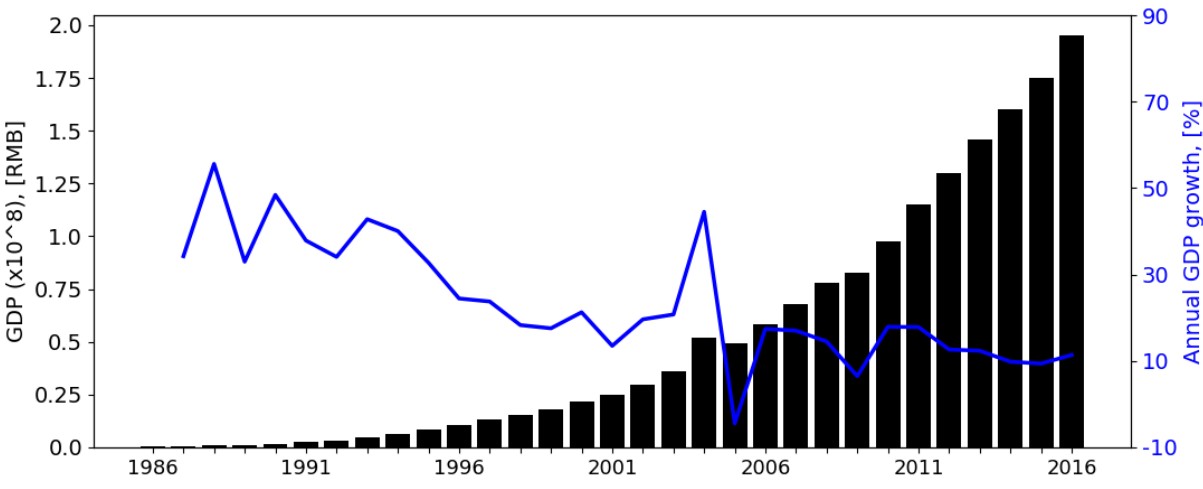

**Figure 2:** The rapid economic growth of Shenzhen, China from 1986–2016. The city's regional GDP (black bar) and annual GDP growth percentage (blue line), i.e., $[(GDP_i - GDP_{i-1}) / GDP_{i-1}]$ x 100% where $i$ = year, are shown.





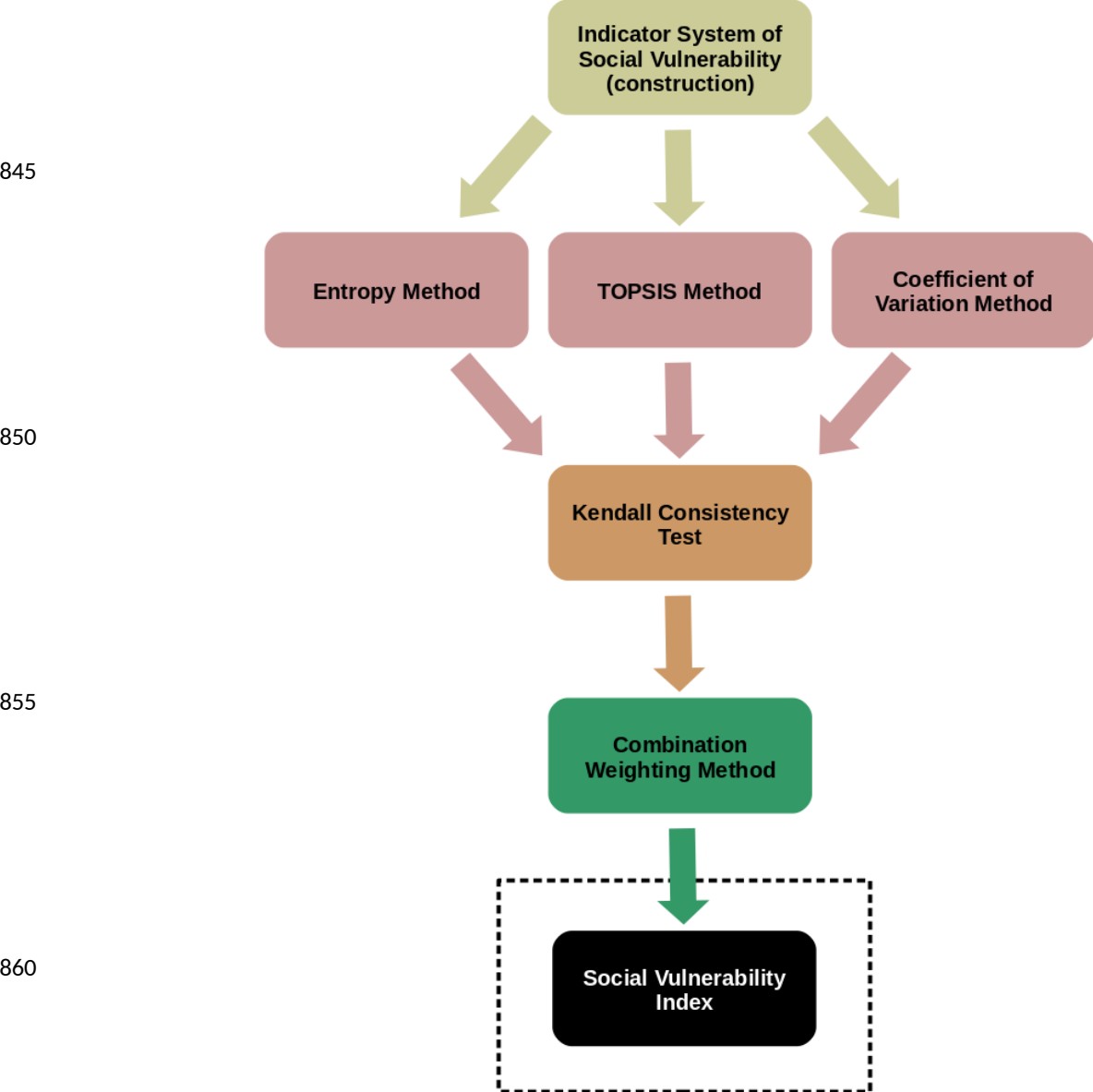




**Figure 3:** Basic four-step procedure (colored boxes) in calculating SVI (black box). The second step (rose boxes) uses three separate methods, while the third (orange box) and fourth (green box) steps are meant to integrate the three calculated results
of the second step. Note, the black dashed box surrounding SVI indicates a result of the four-step process.

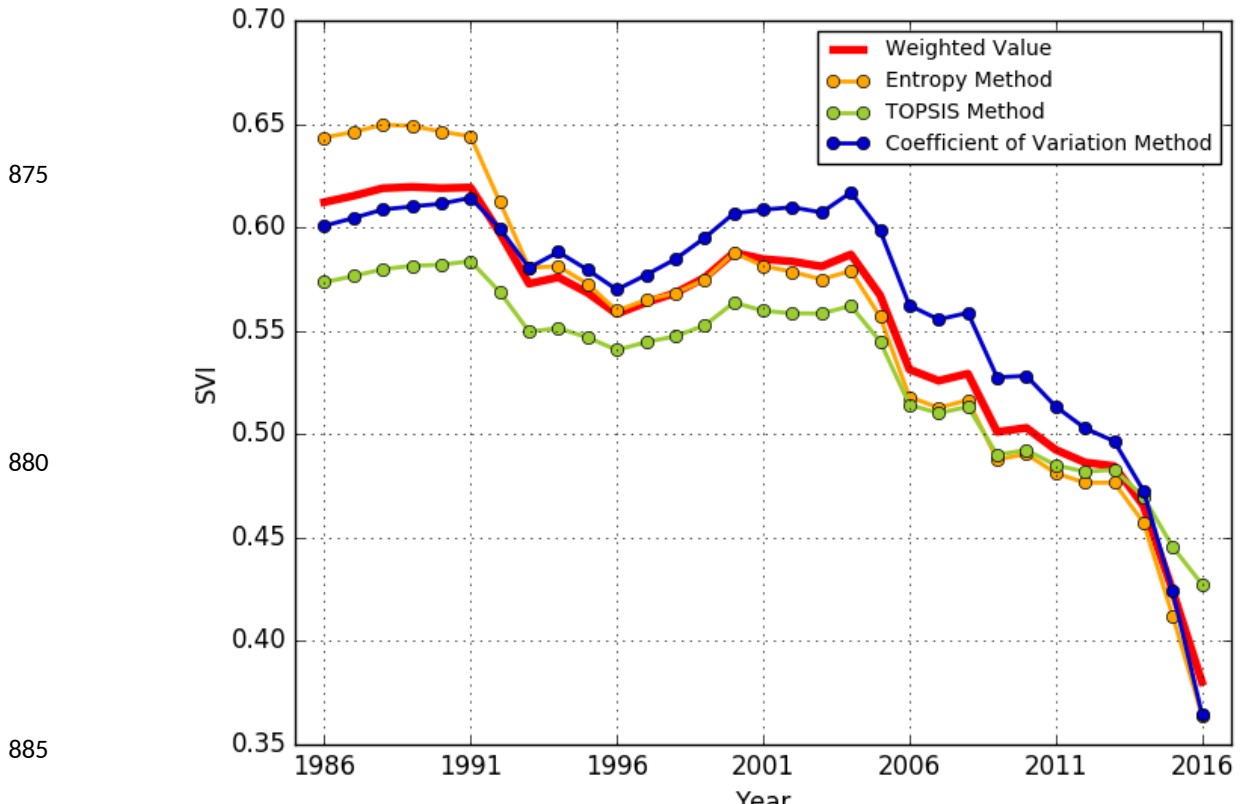

**Figure 4:** SVI aggregated by the Entropy method (yellow line), TOPSIS method (green line) and Coefficient of variation method (blue line). The weighted value of SVI is depicted with thick red line.

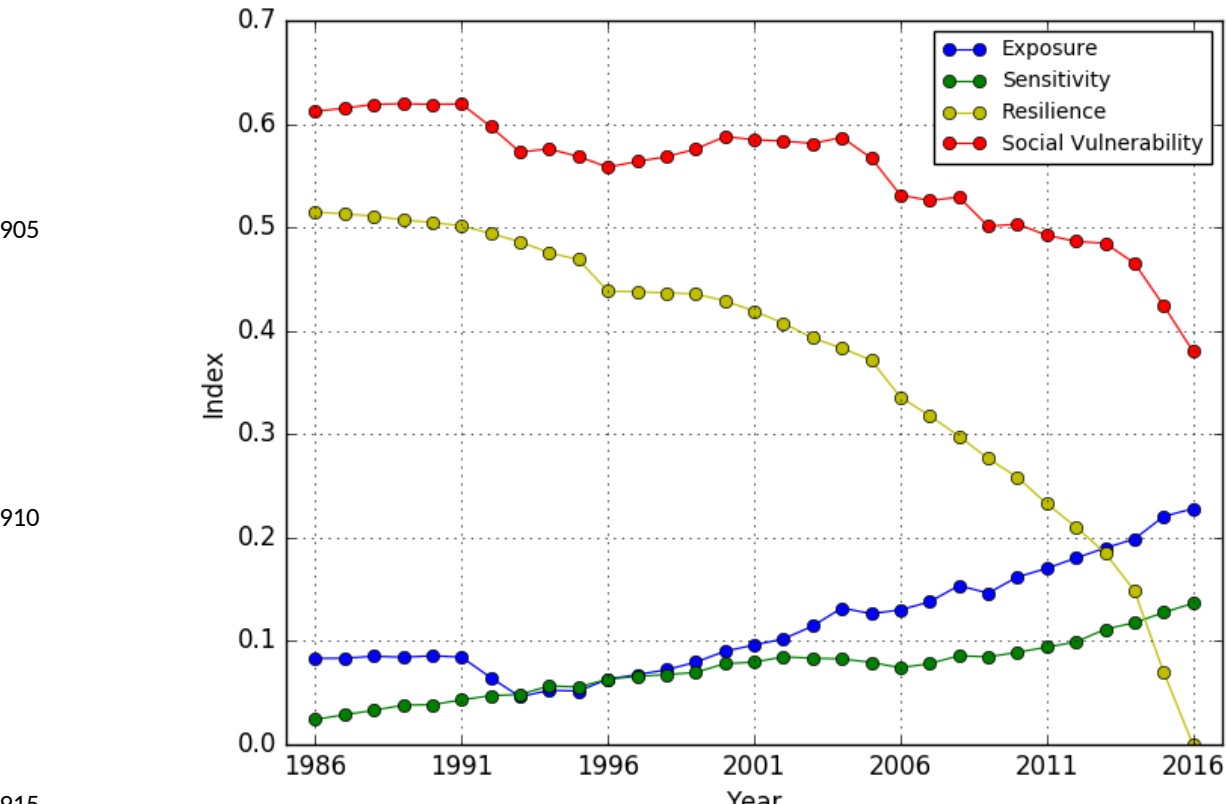

**Figure 5:** Variation of exposure index (EI), sensitivity index (SI) and resilience index (RI). SVI is illustrated in red.


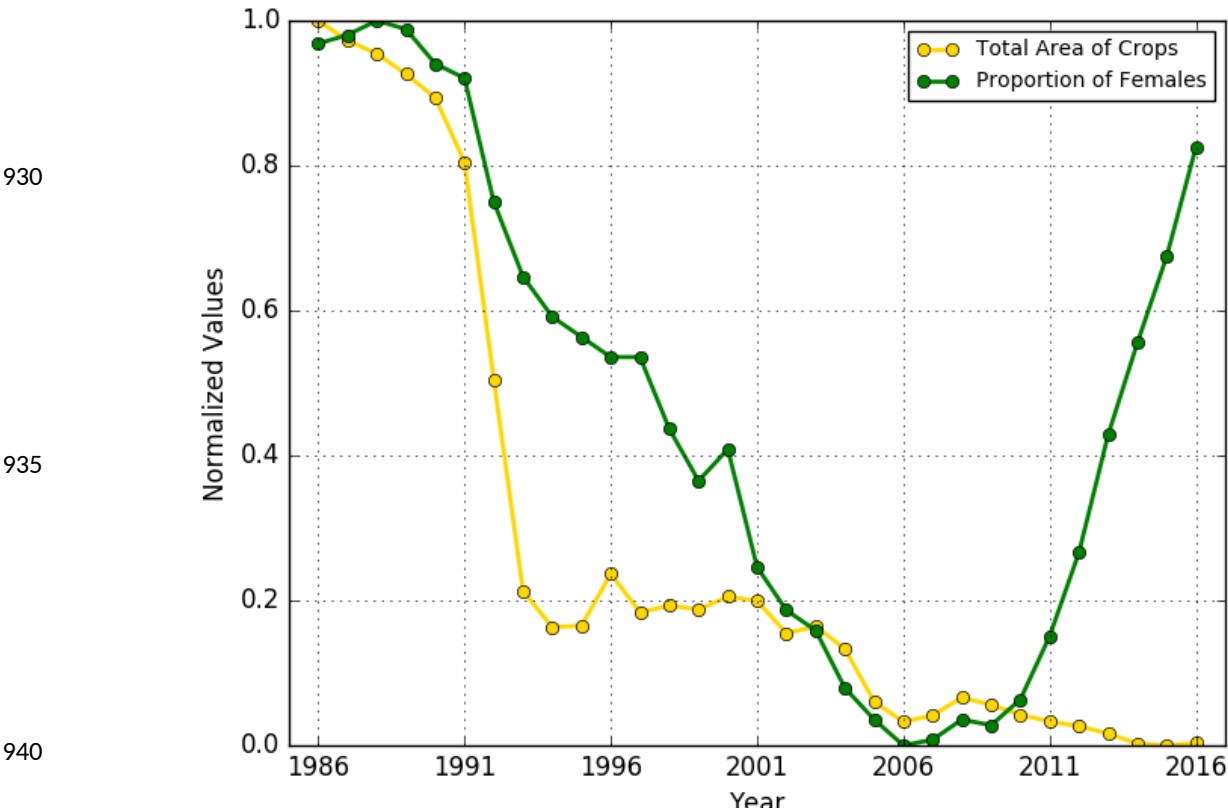




**Figure 6:** Normalized values of total area of crops (yellow line) and proportion of females (green line). Note, the min-max normalization method carries out a linear transformation on the original dataset to standardize each row into an interval [$y_{min}$, 945 $y_{max}$] using the formula: $y = (y_{max} - y_{min})*(x - x_{min}) / (x_{max} - x_{min}) + y_{min}$. These results fall in the interval [0, 1].



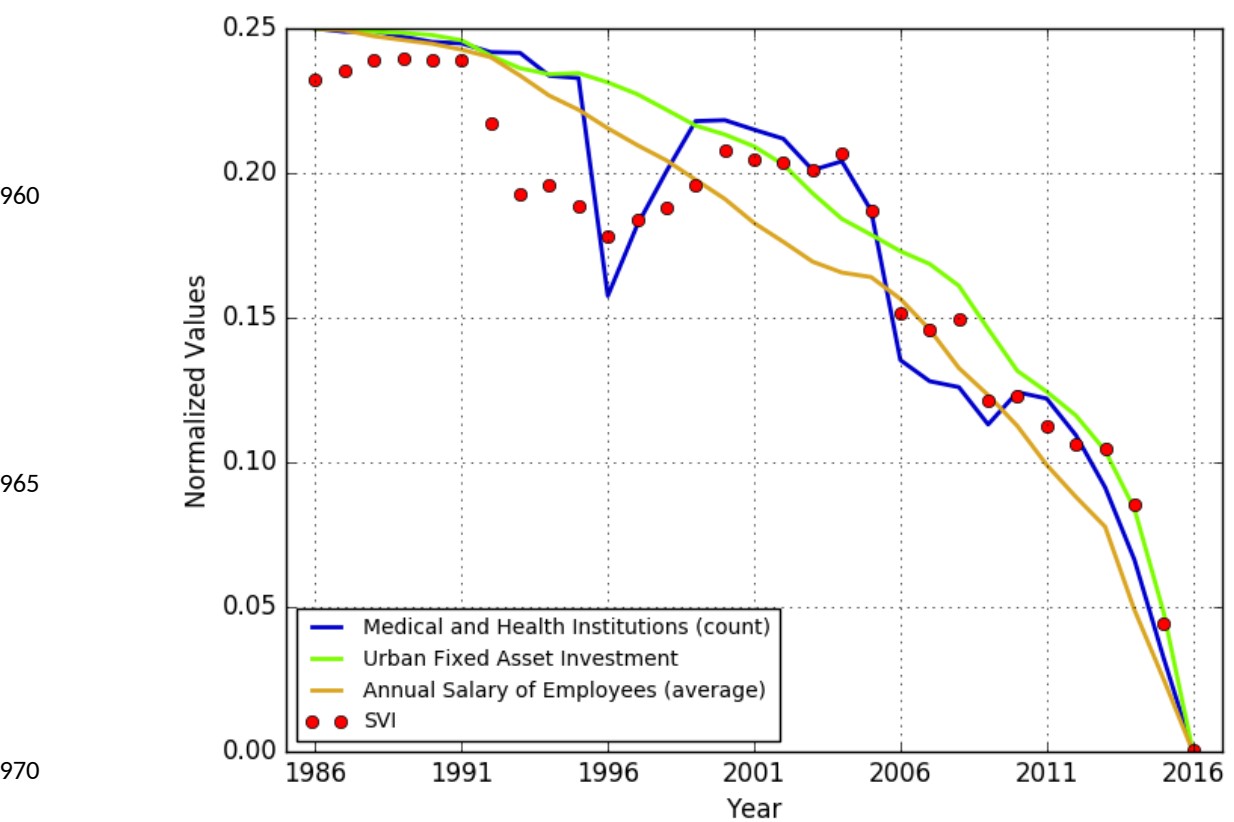




**Figure 7:** Three most relevant indicators of social vulnerability during the research period. SVI is shown in red dots. The min-max normalization method used in **Fig. 6** was used in this figure and the results fall in an interval [0, 0.25]. SVI values were subtracted by a constant (0.38) to meet an identical interval. Note, the y-axis is partially visible to expand the lower portion of the plot.


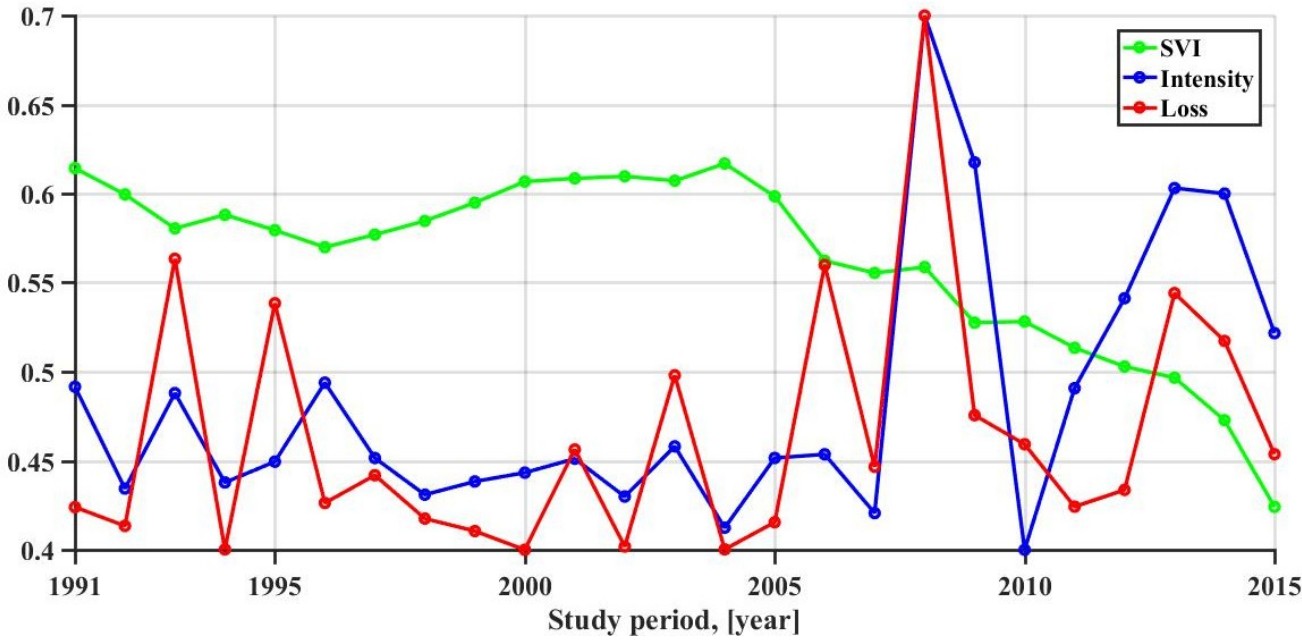

**Figure 8:** Standardized SVI (green line), intensity (blue line) and loss (red line) from 1991 to 2015. Note, (i) the use of min-max normalization and (ii) the range for intensity and loss is unified to the interval [0.4, 0.7] for a convenient comparison with SVI.



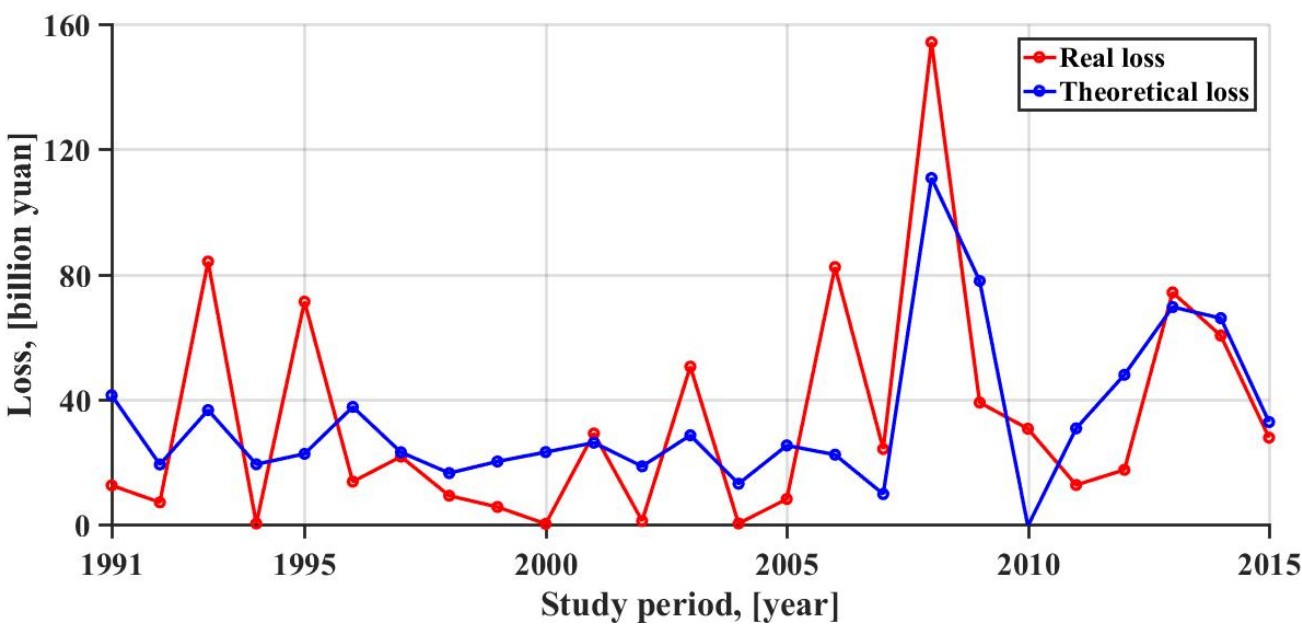

**Figure 9:** Real loss (red line) and theoretical loss (blue line) based on the fitting equation, i.e., loss = 0.01282 + 0.7023*intensity + 0.1986*SVI.


**Table 1:** Indicator system of vulnerability to storm surges in Shenzhen, China.


| Grade I indicators | Grade II indicators |
|---|---|
| Exposure (+) | Permanent resident population at the end of the year (including household and non-household registration) |
| | Regional GDP |
| | Total area of crops |
| | Fishery output value |
| | Port cargo throughput |
| Sensitivity (+) | Gross output value of primary industry |
| | Female proportion |
| | Total enrollment of students |
| | Total social workers at the end of the year |
| Resilience: Per capita (−) | General public budget expenditure |
| | Disposable income of urban residents per capita |
| | Urban fixed asset investment |
| | Average annual salary of employees |
| | Number of medical and health institutions |
| | Number of beds in medical and health institutions |
| | Number of health workers |



35                                               35

**Table 2:** Combined weight coefficients of each single evaluation method.


| | Entropy method | TOPSIS method | Coefficient of variation method |
|---|---|---|---|
| Combined weight coefficient (%) | 42.75 | 25.10 | 32.15 |





**Table 3:** Indicator weight and correlation coefficient of indicator values with SVI.


| Grade I indicators | Grade II indicators | Correlation coefficient with SVI (%) | Indicator weight (%) | |
|---|---|---|---|---|
| **Exposure** **(+)** | Permanent resident population (including household and non-household registration) | −85.48 | 4.13 | 32.05 |
| | Regional GDP | −95.11 | 9.49 | |
| | Total area of crops | 69.92 | 8.33 | |
| | Fishery output value | −40.88 | 3.26 | |
| | Port cargo throughput | −84.39 | 6.84 | |
| **Sensitivity** **(+)** | Gross output value of primary industry | 30.75 | 3.36 | 16.48 |
| | Female proportion | 29.30 | 2.49 | |
| | Total enrollment of students | −89.55 | 6.17 | |
| | Total social workers at the end of the year | −88.69 | 4.45 | |
| **Resilience:** **Per capita** **(−)** | General public budget expenditure | 94.24 | 12.07 | 51.47 |
| | Disposable income of urban residents per capita | 89.85 | 4.99 | |
| | Urban fixed asset investment | 96.31 | 8.00 | |
| | Average annual salary of employees | 95.24 | 6.59 | |
| | Number of medical and health institutions | 97.31 | 6.57 | |
| | Number of beds in medical and health institutions | 95.15 | 6.16 | |
| | Number of health workers | 95.07 | 7.09 | |



