# Peer review of "Trends in social vulnerability to storm surges in Shenzhen, China"

_Natural Hazards and Earth System Sciences, 2019_

## Referee Comment (RC1) · Anonymous Referee #1 · 18 Nov 2019

In this paper, an attempt is made to investigate the temporal variability of social vulnerability to storm surges in Shenzhen, China using the indicator system approach. While the motivation is well established, there are several major issues in this study which the authors need to address.

General comments:

(1) Although the impact of storm surge has been discussed in the introduction, there is no indication of how this study is specifically linked to storm surge. All analyses in this study appear to be based on data of the entire Shenzhen rather than the coastal regions of Shenzhen which are actually vulnerable to storm surge risk. This study appears to be an attempt to quantify the social vulnerability of Shenzhen to all types of natural disaster. Thus the authors might want to reconsider the title of this study.

(2) In this study, the authors did not establish any connection between their social vulnerability index (SVI) and storm surges, i.e. validation of the SVI is not included. For example, Su et al. (2015) used total economic loss of hazards to examine the performance of their SVI which consequently show their SVI is linked to loss due to hazards. The authors could address this problem by relating SVI to economic loss, number of injuries due to storm surges, number of fatalities due to storm surges etc.

(3) There are a lot of statements and claims in this paper but the authors did not include the source of information or the studies to support those statements and claims. (Some of them are listed in the specific comments).

Specific comments:

Page 2, lines 36-38: Reference is needed.

Page 4, lines 121-122: Reference is needed.

Page 6, lines 168-169: Full form of AHP and PCA are needed.

Page 6, lines 175-176, 178: References for these methods would be needed.

Page 6, lines 183-184: What is "a theoretical framework" referring to?

Page 7, line 216, equation 2: What does "lnn" mean?

Page 10, line 289-290: I doubt that Shenzhen faces storm surges accompanied by extratropical cyclones on regular basis. Could you please provide studies to support your statement?

Page 11, lines 314-319: I am not sure about your argument here. Could you please provide studies/evidence that support your claim "students at school and women are more likely to suffer casualties outside" due to the harsh meteorological conditions? Why elderly people and people with disability are not included in the vulnerable groups? Could you please provide studies/evidence that support your claim about "social workers"? Are the authors assuming (all) people would still go to school/work despite a

typhoon is affecting the city? As far as I know, when certain typhoon warning signal (yellow, orange, and red) is issued, school and work would be suspended. People would be asked to stay inside a safe building or evacuate to a safe location, i.e. majority of the people should be in safe locations. Thus I am not sure why students at school, women, and social workers are explicitly included as sensitivity indicators.

Page 12, lines 331-333: Please provide evidence to support your claim – high income level of residents and higher living standard implies strong disaster resilience and faster post-disaster recovery.

Page 12, lines 342-345: It is not clear how does the categorisation of the index, which is developed by Yuan et al. (2016), can be applied to the SVI, which is developed in the current study. These 2 indices do not have the same composition! In addition, the interpretation of this categorisation is not clear. How should we use this categorisation?

Page 13, line 370, 371: Please provide the full form of EI, SI, and RI in the main text.

Page 13, line 374-375: Reference is needed as this information cannot be found in Figure 5.

Figure 2: Figure 2 is not mentioned in the main text! Please name the source(s) of the GDP data set.

Figure 5: It is clear that the downward trend of SVI is mainly driven by the downward trend of RI. However, it is not clear that whether RI could be negative. If RI can be negative, how should it be interpreted? If RI cannot be negative, does this study suggest there exists a threshold in RI that social vulnerability cannot be reduced by further improvement of city's resilience?

Figures 6, 7: I am not sure what does "normalized values" mean.

References:

Su, S. L., Pi, J. H., Wan, C., Li, H. L., Xiao, R., and Li, B. B.: Categorizing so-

cial vulnerability patterns in Chinese coastal cities, Ocean Coast. Manag., 116, 1–8, https://doi.org/10.1016/j.ocecoaman.2015.06.026, 2015.

Yuan, S., Zhao, X., and Li, L. L.: Combination evaluation and case analysis of vulnerability of storm surge in coastal provinces of China, Haiyang Xuebao., 38 (2), 16–24, https://doi.org/10.3969/j.issn.0253-4193.2016.02.002, (in Chinese), 2016.

---

## Referee Comment (RC2) · Anonymous Referee #2 · 22 Dec 2019

General comments:

The proposed research presents an effort to assess the long-term trend in social vulnerability to storm surge induced flood hazard in Shenzhen, China using a system of SV indicators. It was constructed using a complex approach consisting of combination and weighting of results obtained through application of three single evaluation methods. The work is interesting in the context of preparedness, mitigation and adaptation of a large city to natural disaster impacts. The study is well situated among the existing regional assessments and has well defined scope. Authors show fluency in applying and interpreting the risk theory. The number and relevance of proposed indicators are suitable. The assessment of contribution of indicators to final SVI is particularly valuable. Conclusion are very well structured. Such a study is certainly useful

for wide range of stakeholders, especially policy makers, local authorities and coastal managers. Nevertheless, there are several issues that need to be addressed.

Specific comments:

1) As authors themselves pointed out the proposed approach can be used to assess social vulnerability for a variety of disasters, of which they had chosen storm surges. However, detailed enough information on the storm surge induced flood hazard intensity and extent is not provided as well as which coastal areas are most susceptible.

2) Evidently, the lowest level of regional disaggregation is the city of Shenzhen and SVI is relevant for the entire city but not for separate districts within the city limits. This could be a problem with the selected hazard as there is no distinction between areas, which are under direct and indirect hazard impact. Although this is not crucial for a SVI, a better distinction could raise the value of the proposed research. As it is, the study provides only a broad view of the SV in the region. I wonder if there is a possibility to focus on the communities, which are most threatened by flood hazard.

3) I would change "temporal variability to" to "trends in" in the title since a process (variable) can (significantly) vary temporally while SV (despite of minor fluctuations) is more likely to exhibit increasing or decreasing trend over a certain period of time (as discovered by the authors).

4) Compared to the Method chapter, Results & Discussion one seems a bit underdeveloped. I think readers would appreciate a more in depth exposition. The paper's chapters are overly subdivided, which makes the text somewhat choppy. It is suggested to decrease the number of subdivisions.

5) Although the English language used throughout the manuscript is generally correct, it would still need grammar improvements. The text is sometimes rather heavy to follow and understand so stylistic upgrade is also recommendable.

Technical corrections:

A detail list of advisable corrections and specific questions follows. It is solely meant to increase the paper impact by improving the grammar, text fluency and better understanding of the complex connections between methods, resulting indicators and study findings:

L16: Change to "Evaluation of social vulnerability to storm surges is important for any coastal city to provide..."

L19: Change to "which are subsequently combined by weighting in order to calculate a common SVI."

L21-22: Split into 2 sentences

L22: "The research extends further by analysing the city's temporal variability." This sentence is not clear. Temporal variability of what?

L25: Change to "continuous increase of medical services supply"

L26-27: Exposure and sensitivity define vulnerability hence should precede it in the exposition.

L23-24: Change to "during 1986–1991 and 1993–2004... in the rest of the time to form ..."

L27: What do you mean by "causing the changes of social responsibility to transpire"?

Keywords preferably should not repeat the title. Only "Combined evaluation; Indicator system" give additional insight to the study.

L32: Change to "during transition of"

L32-34 Split the first sentence into two sentences

L33: Omit "historical counts of"

L35: Change "ability" to "potential"

L37-38: Provide citations for mentioned catastrophic events

L40: Change to "governments/local authorities managing coastal areas"

L41: Change to "The occurrence of marine natural hazards depends not only on the hazards intensities but also on urban exposure and vulnerability".

L45: Omit "created"

L47: Omit "the risk of". Disaster can be initiated by an event, risk is result of the disaster.

L48: Change "propagate into" to "result in"

L49: Add "In this sense, vulnerability has become..."

L52-53: vulnerability to

L53: Add "reducing the consequences of this type..."

L54: Omit "definition and"

L58: "views about"

L59-61: Rephrase as follows "Based on the theory of sustainable development and from of disaster economics perspective, vulnerability of a system is identified by its ability to prevent and resist a disaster (Turner et al., 2003b)"

L65: Change to "Existing studies divide vulnerability into"

L74: What do you mean by "overall place vulnerability"?

L75 Add "critical" before "infrastructure"

L79: Change to "Before 1990s, ... was paid... were carried out..."

L81: Change to "However, large losses of life and property resulting from the occurrence of more devastating disasters have brought up the attention on the role of social

vulnerability in disaster impact."

L86: Change "management" to "assessment". Management involves not only evaluation.

L86-87: Change to "Hence, governments should analyse . . . policies such as. . . to improve its adaptation capacity. . ."

L88: Change to "considerable amount of research. . . studies on. . ."

L90-91: Change to "Analysis of SV to storm surges . . . is important due to four main reasons"

L91 "Firstly, . . . few assessments of. . . in which . . . is considered"

L92: Choose between "detailed" and "comprehensive". They are synonyms.

L92: What is the object of screening? Could you explain the mechanism of screening?

L92: The process of screening itself cannot be a buffer against disaster.

L94: Change to "other coastal cities, which are exposed to similar or other types of marine natural hazards."

Once more, authors need to specify if and how the methodology was tailored to hazards resulting from storm surges. As it is, it is applicable to any disaster causing floods (and not only), which means the authors should reconsider the title or give additional emphasis on the hazard intensities and extents related to storm surges; in other words, to further justify their choice of disaster.

L95: Change to "Secondly, since 1979, political reform and openness has led . . . in Shenzhen." Omit "during the study period"

L95: Omit "expedited process"; "rapid" or "accelerated" is enough to describe the process.

L97: Change to "By choosing Shenzhen, we study a typical scenario of SV change as

a result of . . ." L98: "Thirdly, so far, . . ."

L100: "Instead, herewith, a composite. . ."

L102: "Data envelopment analysis (DEA). . .evaluation in China to discover . . ."

L105-106: Rephrase "Five methods for combined evaluation were used by Liu and Liu (2017) and results determined that among seven coastal cities selected for evaluation in Shandong Province Yantai city and Binzhou city had the highest and lowest vulnerability, respectively."

L108: Rephrase "The socioeconomic vulnerability to typhoon-induced storm surges for municipal districts of Guangdong Province was assessed using the fuzzy comprehensive evaluation. It was determined that vulnerability presented a large spatial heterogeneity (Zhang et al., 2010)."

L111: Omit "with results from Zhang et al. (2010)"

L112: Change "differences" to "dimensions".

L115: Temporal patterns = trend ?

L116: Can you explain what you mean by "macroscopic angle"?

Here, you can state again the period your research covers.

L122: Change to "Since its establishment in 1979, in just 40 years, . . . through a . . ."

L124: Change to "However, due to its location at the coast of the Pearl River Delta (Fig. 1a,b) and its proximity to the northern part of the South China Sea (Fig. 1b,c), Shenzhen is facing many coastal disasters threatening its sustainable development, among which storm surge induced disasters are the most severe."

L127: "[http://www.sz.gov.cn/ytqzfzx/yingji/yjya/201712/t20171206_10111758.htm (last access: 30 June 2019)]" This should be moved to the list of references.

L129: "116 typhoons have seriously affected the Shenzhen coastal area"

L136: Maybe you mean "The increased frequency of storm surges"

L137-138: It is not clear if EWS is a recommendation? Can you relate this to your research results? Which are those "particularly susceptible areas"?

L140: Change "fully contained" to "entirely available"

L141-142: Change "which was compiled by the Shenzhen Statistical Bureau and a Shenzhen-based investigation team of the National Bureau of Statistics, and published (updated annually) by the Shenzhen Statistical Bureau." to "which is compiled and published on annual basis".

L153-154: Change to "Due to the absence of long-term statistical data on some important indicators, this study is limited to a partial statistical dataset spanning the period 1986 - 2016 in order to sustain the data integrity."

L165: By "evaluation result", don't you mean "final score"?

L175-178: Provide citations for all mentioned methods and tests.

L179&201: Method or strategy?

L 179-181: Rephrase as follows "Finally, the combined evaluation results are achieved, which have significant advantages compared to those of all single methods due to weighted value of each evaluation method."

L183-184: It is not clear if the first sentence describe a statement or research action.

L187: I cannot understand what the meaning of "knowledge simplicity attribute of rough set" is.

L188: Among which?

L199: Change "all above evaluation methods" to "all evaluation methods in use"

L201: Using "a single evaluation framework", do you refer to combined weighting method?

L213: Change to "Calculate the proportion of the indicator j in year i (rij).

L234: The meaning of the second sentence in the paragraph is not clear.

L260: Due to limitations of the methods in use, each single evaluation can lead to a different conclusion."

L260: Use "Nevertheless" instead of "However".

Section 2.4 is really hard to follow. Could you better explain the connection between methods (section 2.3) and resulting indicators (section 2.4)?

L286-288: It is not clear if this is achieved within previous researches or is a stage within the present study.

L289: Storm surges are caused by the action of tropical and extratropical cyclones not accompanied by them.

L290: Could you explain better the screening process? What is the reason to take out of consideration man-made barriers?

L291: To what disaster body do you refer: the city itself or a body in general? Which are the bodies you screen?

L291-293: The sentences seem to repeat one another. Could you clarify and rephrase.

L304: Change "While regional GDP" to "Since the amount of regional GDP"

L305: Change "equates" to "corresponds", "for" to "to".

The last two sentences in section 2.4.1 should change their places.

L312: Again, is Shenzhen city the disaster body? Industries of primary importance for Shenzhen city are. . .

L313: Change "fluctuations" to "changes"

L315: Change "higher winds and precipitation patterns" to "severe winds and precipitations"

L316: Change "inconvenient" to "busy"

L317: Change "suffer casualties outside" to "suffer injuries or even cause casualties"

L322: Change "with which" to "meaning that"

L324: Change "aspects" to "groups"

L326: Change "more money is devoted into" to "more resources are provided/spent for"

L330: Add "consequences" after "resists disaster"

L331: "per capita"

L333: Change "infrastructure construction" to "public services"

L334-335: Change to "level of medical and health care, including the number of medical and health institutions and their equipment (e.g. beds etc. . .) as well as the number of health employees."

L336: "potential victims"

L342: Change "social vulnerability to storm surges discussed in this research can be approximately divided into" to "degrees of social vulnerability to storm surges discussed in this research are set to. . ."

L345: It is not clear if proposed SVI threshold values were calculated by the authors or were borrowed from Yuan et al. (2016).

L346: Change "close" to "similar"

L422: Change "obvious" to "pronounced", "variation" – "variability"

Figure 2 is not referred to in the text.

L753: Change to "and outlined using crimson colour."

L836: Change to "method (blue line). The weighted value of SVI is depicted with thick red line."

Table1/3: Resilience: Per capita
* * *

---

## Author Comment (AC1) · 12 Feb 2020

**Response to Anonymous Reviewer #1**

The research group thanks Anonymous Reviewer #1 for their detailed comments and supportive suggestions. This feedback allows our group to make appropriate updates to the manuscript in preparation for uploading an improved version. Our answers to the General comments and Specific comments are as follows (denoted with a > symbol and blue text):

General comments:

(1) Although the impact of storm surge has been discussed in the introduction, there is no indication of how this study is specifically linked to storm surge. All analyses in this study appear to be based on data of the entire Shenzhen rather than the coastal regions of Shenzhen which are actually vulnerable to storm surge risk. This study appears to be an attempt to quantify the social vulnerability of Shenzhen to all types of natural disaster. Thus the authors might want to reconsider the title of this study.

> Yes, it is a good observation about this paper. Due to a data acquisition limitation, it is very difficult or impossible to find perfect indicators with a long-term record specifically linked to storm surge in China. This work is a creative attempt to analyze publicly available "macroscopic" data in order to explain the "microscopic" phenomena for such similar Chinese coastal cities. Furthermore, the data's spatial coverage is a narrow, 20 km length in the north-south direction across Shenzhen City and therefore most areas of Shenzhen are threatened during storm surges. Additionally, some directly related indicators of storm surge are selected, rather than other types of natural disasters such as "fishery output value" and "port cargo throughput", and used in the indicator system for evaluating social vulnerability. We believe our results address the problem to a certain extent. From this research, it becomes feasible for us to deliver suggestions to local governments about the need to collect and archive statistical data for most threatened coastal communities. Also, we will make an appropriate title change to fit this study more closely.

(2) In this study, the authors did not establish any connection between their social vulnerability index (SVI) and storm surges, i.e. validation of the SVI is not included. For example, Su et al. (2015) used total economic loss of hazards to examine the performance of their SVI which consequently show their SVI is linked to loss due to hazards. The authors could address this problem by relating SVI to economic loss, number of injuries due to storm surges, number of fatalities due to storm surges etc.

> Yes, this is a very constructive idea. We considered adding a validation strategy but public data of economic loss due to storm surges in Shenzhen is simply unavailable. Economic loss data was found to be available for Guangdong province (i.e. a broader scale dataset) which is where Shenzhen is located. We assume that the loss from a storm surge disaster in Shenzhen and Guangdong province are in direct proportion. We will add a validation section to this manuscript.

(3) There are a lot of statements and claims in this paper but the authors did not include the source of information or the studies to support those statements and claims. (Some of them are listed in the specific comments).

> Yes, thank you for your reminder. We will add all necessary sources of information to support our statements and claims, while satisfying your comment.

Specific comments (Technical corrections):

Page 2, lines 36-38: Reference is needed.

> The following citations account for the mentioned catastrophic events within lines 36-38 and were added to the manuscript:

Forbes, C., Rhome, J., Mattocks, C., and Taylor, A. A.: Predicting the storm surge threat of Hurricane Sandy with the National Weather Service SLOSH model, J. Mar. Sci. Eng., 2 (2), 437–476, https://doi.org/10.3390/jmse2020437, 2014.

Frank, N. L., and Husain, S. A.: The deadliest cyclone in history? Bull. Am. Meteor. Soc., 52 (6), 438–445, 1971.

Fritz, H. M., Blount, C., Sokoloski, R., Singleton, J., Fuggle, A., McAdoo, B. G., Moore, A., Grass, C., and Tate, B.: Hurricane Katrina storm surge distribution and field observations on the Mississippi Barrier Islands, Estuar. Coast. Shelf Sci., 74 (1-2), 12–20, https://doi.org/10.1016/j.ecss.2007.03.015, 2007.

Fritz, H. M., Blount, C., Thwin, S., Thu, M. K., and Chan, N.: Cyclone Nargis storm surge in Myanmar, Nature Geosci., 2 (7), 448–449, https://doi.org/10.1038/ngeo558, 2009.

Irish, J. L., Resio, D. T., and Ratcliff, J. J.: The influence of storm size on hurricane surge, J. Phys. Oceanogr., 38 (9), 2003–2013, https://doi.org/10.1175/2008JPO3727.1, 2008.

Lagmay, A. M. F, Agaton, R. P., Bahala, M. A. C., Briones, J. B. L. T., Cabacaba, K. M. C., Caro, C. V. C., Dasallas, L. L., Gonzalo, L. A. L., Ladiero, C. N., Lapidez, J. P., Mungcal, M. T. F., Puno, J. V. R., Ramos, M. M. A. C., Santiago, J., Suarez, J. K., and Tablazon, J. P.: Devastating storm surges of Typhoon Haiyan, Int. J. Disast. Risk Re., 11, 1–12, https://doi.org/10.1016/j.ijdrr.2014.10.006, 2015.

Rosenzweig, C., and Solecki, W.: Hurricane Sandy and adaptation pathways in New York: Lessons from a first-responder city, Glob. Environ. Change, 28, 395–408, https://doi.org/10.1016/j.gloenvcha.2014.05.003, 2014.

Xian, S., Feng, K., Lin, N., Marsooli, R., Chavas, D., Chen, J., and Hatzikyriakou, A.: Brief communication: Rapid assessment of damaged residential buildings in the Florida Keys after Hurricane Irma, Nat. Hazards Earth Syst. Sci., 18, 2041–2045, https://doi.org/10.5194/nhess-18-2041-2018, 2018.

Yi, C. J., Suppasri, A., Kure, S., Bricker, J. D., Mas, E., Quimpo, M., and Yasuda, M.: Storm surge mapping of typhoon Haiyan and its impact in Tanauan, Leyte, Philippines, Int. J. Disast. Risk Re., 13, 207–214, https://doi.org/10.1016/j.ijdrr.2015.05.007, 2015.

Page 4, lines 121-122: Reference is needed.

> The following citation accounts for the content about Shenzhen's GDP on line 121-122:

Zünd, D. and Bettencourt, L. M. A.: Growth and development in prefecture-level cities in China, PLoS ONE, 14(9), e0221017, https://doi.org/10.1371/journal.pone.0221017, 2019.

Also, a change from "attributed to the highest per capita Gross Domestic Product (GDP) in mainland China" to "attributed to one of the highest Gross Domestic Product (GDP) per capita in mainland

China..." due to the current GDP ranking, i.e. 1. Shanghai 2. Beijing 3. Shenzhen 4. Tianjin. The small city of Ordos (rich in natural resources) is noted as being the leader in the highest GDP per capita in mainland China although its significantly smaller in geographic size.

Page 6, lines 168-169: Full form of AHP and PCA are needed.

> The terms analytic hierarchy process and principal component analysis were added to the text.

Page 6, lines 175-176, 178: References for these methods would be needed.

> Replace the words with "Firstly, the construction of an optimized social vulnerability evaluation indicator system, based on the idea of rough set theory (Das et al., 2018), is completed. Second, the entropy method (Zhou and Yang, 2019), the Technique for Order Preference by Similarity to an Ideal Solution (TOPSIS) method (Kuo, 2017) and the coefficient of variation method (Zhou et al., 2004) are used to weigh the indicators and aggregate SVI separately. Then, the consistency of different evaluation results is tested by using the compatibility test method, i.e., Kendall consistency test (Wen and Hu, 2002)." All of the references have been included already.

Page 6, lines 183-184: What is "a theoretical framework" referring to?

> "a theoretical framework" was changed to "vulnerability theory"

Page 7, line 216, equation 2: What does "lnn" mean?

> Equation (2) was changed to $e_{jitk}$
In case you cannot see the formula clearly, a larger picture is shown below.

$$e_j = -(\ln n)^{-1} \sum_{i=1}^{n} \bar{r}_{ij} \ln \bar{r}_{ij} \, (0 \le e_j \le 1, j = 1,2,3,\cdots,m)$$

Page 10, line 289-290: I doubt that Shenzhen faces storm surges accompanied by extratropical cyclones on regular basis. Could you please provide studies to support your statement?

> Thank you for pointing out this mistake. It is an error and we deleted "extratropical cyclones" from the manuscript, i.e. line 289. Storm surges that affect Shenzhen are mainly caused by tropical cyclones.

Page 11, lines 314-319: I am not sure about your argument here. Could you please provide studies/evidence that support your claim "students at school and women are more likely to suffer casualties outside" due to the harsh meteorological conditions? Why elderly people and people with disability are not included in the vulnerable groups? Could you please provide studies/evidence that support your claim about "social workers"? Are the authors assuming (all) people would still go to school/work despite a typhoon is affecting the city? As far as I know, when certain typhoon warning signal (yellow, orange, and red) is issued, school and work would be suspended. People would be asked to stay inside a safe building or evacuate to a safe location, i.e. majority of the people should be in safe locations. Thus I am not sure why students at school, women, and social workers are explicitly included as sensitivity indicators.

> Thanks for your detailed observation and relevant questions. With regard to the sensitivity indicators, as we know, the occurrence of storm surges is uncertain and the early warning system is not that accurate. Unfortunately, when a strong storm surge occurrence happens, it is difficult to have all students, women and social workers be held in the safe place. Additionally, elderly people and people with disabilities are included in vulnerable groups (Yuan et al., 2016), but there is no specific data captured about the elderly population in Shenzhen's statistical yearbooks. We would like to include all factors that would reach the general agreement of the marine disaster community. However, due to the lack of original data, we can only provide certain factors to the indicator system for analysis.

Also we will add the following reference (already in the reference list) to line 317:

Yuan, S., Zhao, X., and Li, L. L.: Combination evaluation and case analysis of vulnerability of storm surge in coastal provinces of China, Haiyang Xuebao., 38 (2), 16–24, https://doi.org/10.3969/j.issn.0253-4193.2016.02.002, (in Chinese), 2016.

Page 12, lines 331-333: Please provide evidence to support your claim – high income level of residents and higher living standard implies strong disaster resilience and faster post-disaster recovery.

> We will add the following reference (already in the reference list) to line 333:

Yuan, S., Zhao, X., and Li, L. L.: Combination evaluation and case analysis of vulnerability of storm surge in coastal provinces of China, Haiyang Xuebao., 38 (2), 16–24, https://doi.org/10.3969/j.issn.0253-4193.2016.02.002, (in Chinese), 2016.

Page 12, lines 342-345: It is not clear how does the categorisation of the index, which is developed by Yuan et al. (2016), can be applied to the SVI, which is developed in the current study. These 2 indices do not have the same composition! In addition, the interpretation of this categorisation is not clear. How should we use this categorisation?

> Thanks for pointing out this important mistake. We neglected the difference between the two studies and we have adjusted the categorisation in the updated manuscript.

Regarding lines 342-345:

Change the sentence "According to previous studies on disaster vulnerability, social vulnerability to storm surges discussed in this research can be approximately divided into (i) high vulnerability, (ii) relatively high vulnerability, (iii) moderate vulnerability, (iv) relatively low vulnerability and (v) low vulnerability and the corresponding critical points of SVI are 0.5873, 0.5163, 0.4452 and 0.3741, respectively (Yuan et al., 2016)." to "According to the common idea of equal division in mathematical statistics, social vulnerability to storm surges discussed in this research can be approximately divided into (i) high vulnerability, (ii) relatively high vulnerability, (iii) moderate vulnerability, (iv) relatively low vulnerability and (v) low vulnerability and the corresponding critical points of SVI are 0.5715, 0.5237, 0.4759 and 0.4281, respectively."

Regarding lines 355-358:

Change the sentence "According to classification criteria, social vulnerability to storm surges in Shenzhen during the entire study period can be divided into four stages: (i) high social vulnerability between 1986 to 1992, (ii) relatively high social vulnerability between 1993 to 2008, (iii) moderate

social vulnerability between 2009 and 2014, and (iv) relatively low social vulnerability between 2015 and 2016. The time to maintain relatively high (low) social vulnerability is the longest (shortest) as a whole, respectively." to "According to classification criteria, social vulnerability to storm surges in Shenzhen during the entire study period can be divided into five stages: (i) high social vulnerability between 1986 to 1994 and 1999 to 2004, (ii) relatively high social vulnerability between 1995 to 1998 and 2005 to 2008, (iii) moderate social vulnerability between 2009 to 2013, (iv) relatively low social vulnerability in 2014 and (v) low vulnerability in 2015 and 2016. The time to maintain high social vulnerability is the longest and relatively low social vulnerability is the shortest as a whole, respectively. It is apparent that, after 2008, social vulnerability has been completely removed from relatively high levels."

Page 13, line 370, 371: Please provide the full form of EI, SI, and RI in the main text.

> The terms exposure index, sensitivity index and resilience index were added to the text.

Page 13, line 374-375: Reference is needed as this information cannot be found in Figure 5.

> We understand the reviewer's concern. Fiscal spending, residents' income levels, completion degree of medical conditions, and infrastructure are all included in the indicator system and they all belong to resilience indicators. In Figure 5, we can see the continuous decrease of RI and that is caused by the improvement of Shenzhen's fiscal spending, residents' income levels, completion degree of medical conditions, and infrastructure. While the conclusion cannot be found in Figure 5, it can be indicated by the trend of RI. Therefore, we believe a reference is not needed here.

Figure 2: Figure 2 is not mentioned in the main text! Please name the source(s) of the GDP data set.

> Add "Through the growth of GDP, it is found that Shenzhen's economic level is progressively advancing during our study period (Fig. 2)." at the end of the first paragraph in Section **2.1 Study area**. The source of the GDP data set is the same as Section **2.2 Data sources**.

Figure 5: It is clear that the downward trend of SVI is mainly driven by the downward trend of RI. However, it is not clear that whether RI could be negative. If RI can be negative, how should it be interpreted? If RI cannot be negative, does this study suggest there exists a threshold in RI that social vulnerability cannot be reduced by further improvement of city's resilience?

> It's true that the downward trend of SVI is mainly driven by the downward trend of RI, but RI cannot be negative. In 2016, all of the resilience indicators happen to reach their maximum, and when performing normalization, the resilience index in 2016 equals zero (zero is the minimum of RI). Social vulnerability can be reduced by further improvement of the city's resilience because when RI equals zero, it only indicates that the resilience for that year is the strongest compared with other years during the whole study period, rather than the city improving its resilience to the largest degree.

Figures 6, 7: I am not sure what does "normalized values" mean.

> The graph objects are all normalized to a uniform range for comparison. Since their dimensions are different, they are uniformly converted into dimensionless quantities. In figure 6, we use min-max normalization which means a linear transformation is performed on the original data so that the result falls into the interval [0,1]. In figure 7, we also use min-max normalization but the value of indicators

fall into the interval [0,0.25]. For SVI, we subtract 0.38 from the value to yield an interval [0,0.25]. As a result, we can easily compare the relationship between variables.

\*\*\*\*\*\*\*\*\*\*\*\*\*\*\*\*\*\*\*\*\*\*\*\*\*\*\*\*\*\*\*\*\*\*\*\*\*\*\*\*\*\*\*\*\*\*\*\*\*\*\*\*\*\*\*\*\*\*\*\*\*\*\*\*\*\*\*\*\*\*\*\*\*\*\*\*\*\*

References used by Anonymous Reviewer #1:

Su, S. L., Pi, J. H., Wan, C., Li, H. L., Xiao, R., and Li, B. B.: Categorizing social vulnerability patterns in Chinese coastal cities, Ocean Coast. Manag., 116, 1–8, https://doi.org/10.1016/j.ocecoaman.2015.06.026, 2015.

Yuan, S., Zhao, X., and Li, L. L.: Combination evaluation and case analysis of vulnerability of storm surge in coastal provinces of China, Haiyang Xuebao., 38 (2), 16–24, https://doi.org/10.3969/j.issn.0253-4193.2016.02.002, (in Chinese), 2016.

---

## Author Comment (AC2) · 13 Feb 2020

**Response to Anonymous Reviewer #2**

The research group thanks Anonymous Reviewer #2 for their careful review, constructive feedback and technical screening. We know the reviewer's comments, suggestions and list of technical corrections will allow our group to forge an enhanced version of the manuscript. Our answers to the specific comments and technical corrections are as follows (denoted with a > symbol and blue text). Some additions are placed in magenta text.

**General comments:**

The proposed research presents an effort to assess the long-term trend in social vulnerability to storm surge induced flood hazard in Shenzhen, China using a system of SV indicators. It was constructed using a complex approach consisting of combination and weighting of results obtained through application of three single evaluation methods. The work is interesting in the context of preparedness, mitigation and adaptation of a large city to natural disaster impacts. The study is well situated among the existing regional assessments and has well defined scope. Authors show fluency in applying and interpreting the risk theory. The number and relevance of proposed indicators are suitable. The assessment of contribution of indicators to final SVI is particularly valuable. Conclusion are very well structured. Such a study is certainly useful for wide range of stakeholders, especially policy makers, local authorities and coastal managers. Nevertheless, there are several issues that need to be addressed.

**Specific comments:**

(1) As authors themselves pointed out the proposed approach can be used to assess social vulnerability for a variety of disasters, of which they had chosen storm surges. However, detailed enough information on the storm surge induced flood hazard intensity and extent is not provided as well as which coastal areas are most susceptible.

> Yes, detailed information on storm surge-induced flood hazard intensity and extent is provided. Information about coastal areas, which are most susceptible, is located in the first paragraph of the **Introduction**.

(2) Evidently, the lowest level of regional disaggregation is the city of Shenzhen and SVI is relevant for the entire city but not for separate districts within the city limits. This could be a problem with the selected hazard as there is no distinction between areas,which are under direct and indirect hazard impact. Although this is not crucial for a SVI, a better distinction could raise the value of the proposed research. As it is, the study provides only a broad view of the SV in the region. I wonder if there is a possibility to focus on the communities, which are most threatened by flood hazard.

> Yes, we understand your concern. We'd like to focus on small scale districts and communities which are most threatened by storm surge but it is difficult or even impossible to obtain community data at this spatial resolution. This work is a creative attempt to analyze publicly available "macroscopic" data in order to explain the "microscopic" phenomena for such similar Chinese coastal cities. Furthermore, the data's spatial coverage is a narrow, 20 km length in the north-south direction across Shenzhen City and therefore most areas of Shenzhen are threatened during storm surges. We believe our results address the problem to a certain extent. From this research, it becomes feasible for us to deliver suggestions to local governments about the need to collect and archive statistical data for most threatened coastal communities.

(3) I would change "temporal variability to" to "trends in" in the title since a process (variable) can (significantly) vary temporally while SV (despite of minor fluctuations) is more likely to exhibit increasing or decreasing trend over a certain period of time (as discovered by the authors).

> Yes, it is a very good suggestion. We made a title change from "temporal variability to" to "trends in" in the updated manuscript.

(4) Compared to the Method chapter, Results & Discussion one seems a bit under developed.  I think readers would appreciate a more in depth exposition.  The paper's chapters are overly subdivided, which makes the text somewhat choppy.  It is suggested to decrease the number of subdivisions.

> Yes, you are correct and we agree with you.  We will condense the chapters to keep only secondary titles, especially, the Method chapter.  The Results & Discussion chapter will be balanced (content wise) with the other chapters, so research results are expressed in a more holistic manner.

(5) Although the English language used throughout the manuscript is generally correct, it would still need grammar improvements.  The text is sometimes rather heavy to follow and understand so stylistic upgrade is also recommendable.

> Yes, we do agree with you.  In the updated manuscript, the grammar and English language will be improved.  We will work on reducing the heaviness of the text and balance the content better for readability purposes.

**Technical corrections:**

A detail list of advisable corrections and specific questions follows. It is solely meant to increase the paper impact by improving the grammar, text fluency and better understanding of the complex connections between methods, resulting indicators and study findings:

L16: Change to "Evaluation of social vulnerability to storm surges is important for any coastal city to provide..."

> Changed to "Evaluation of social vulnerability to storm surges is important for any coastal city to provide..."

L19: Change to "which are subsequently combined by weighting in order to calculate a common SVI."

> Changed to "which are subsequently combined by weighting in order to calculate a common SVI."

L21-22: Split into 2 sentences

> The sentence was split into 2 sentences and now reads "Shenzhen has a current reputation of having the most economic development potential and is a representative city in China.  The city is chosen to evaluate its social vulnerability to storm surges via a historical social and economic statistical dataset spanning from 1986 to 2016."

L22: "The research extends further by analysing the city's temporal variability." This sentence is not clear. Temporal variability of what?

> This sentence "The research extends further by analysing the city's temporal variability." was deleted.

L25: Change to "continuous increase of medical services supply"

> Changed to "continuous increase of medical services supply"

L26-27: Exposure and sensitivity define vulnerability hence should precede it in the exposition.

> Changed from "Results reveal that social vulnerability keeps almost constant from 1986–1991 and 1993–2004, while it decreased sharply in the remainder of times to show a 'stair-type' declining curve over the past 30 years.  Resilience is progressively increasing by virtue of a continuous increase in medical institutions, fixed asset investments and salary levels of employees.  These determinants contribute to the overall downward trend of social vulnerability for Shenzhen.  Exposure and sensitivity increased slowly with some fluctuation, causing the changes of social responsibility to transpire."
to
"Results reveal that resilience is progressively increasing by virtue of a continuous increase in medical institutions, fixed asset investments and salary levels of employees.  These determinants contribute to the overall downward trend of social vulnerability for Shenzhen.  Exposure and sensitivity increased slowly with some fluctuation, leading to fluctuations in the social vulnerability results.  Social vulnerability keeps constant from 1986–1991 and 1993–2004, while it decreased sharply in the remainder of times to show a 'stair-type' declining curve over the past 30 years."

L23-24: Change to "during 1986–1991 and 1993–2004… in the rest of the time to form..."

> Changed to "during 1986–1991 and 1993–2004… in the rest of the time to form..."

L27: What do you mean by "causing the changes of social responsibility to transpire"?

> Changed from "causing the changes of social vulnerability to transpire." to "leading to fluctuations in the trends of social vulnerability."

Keywords preferably should not repeat the title. Only "Combined evaluation; Indicator system" give additional insight to the study.

> Yes, we will keep "Combined evaluation; Indicator system" as keywords and include other keywords.

L32: Change to "during transition of"

> Changed to "during transition of"

L32-34 Split the first sentence into two sentences

> The sentence was split into two sentences and reads as "Storm surge refers to the abnormal volumetric rise of sea water layered above the astronomical tide due to severe meteorological

conditions experienced during transition of low-pressure weather systems.  Tropical and extratropical cyclones rank near the pinnacle among marine natural hazards in terms of human casualties and expensive infrastructure losses."

L33: Omit "historical counts of"

> "historical counts of" was omitted from L33.

L35: Change "ability" to "potential"

> Changed from "ability" to "potential" in the updated manuscript.

L37-38: Provide citations for mentioned catastrophic events

> Forbes, C., Rhome, J., Mattocks, C., and Taylor, A. A.: Predicting the storm surge threat of Hurricane Sandy with the National Weather Service SLOSH model, J. Mar. Sci. Eng., 2 (2), 437–476, https://doi.org/10.3390/jmse2020437, 2014.

Frank, N. L., and Husain, S. A.: The deadliest cyclone in history? Bull. Am. Meteor. Soc., 52 (6), 438–445, 1971.

Fritz, H. M., Blount, C., Sokoloski, R., Singleton, J., Fuggle, A., McAdoo, B. G., Moore, A., Grass, C., and Tate, B.: Hurricane Katrina storm surge distribution and field observations on the Mississippi Barrier Islands, Estuar. Coast. Shelf Sci., 74 (1-2), 12–20, https://doi.org/10.1016/j.ecss.2007.03.015, 2007.

Fritz, H. M., Blount, C., Thwin, S., Thu, M. K., and Chan, N.: Cyclone Nargis storm surge in Myanmar, Nature Geosci., 2 (7), 448–449, https://doi.org/10.1038/ngeo558, 2009.

Irish, J. L., Resio, D. T., and Ratcliff, J. J.: The influence of storm size on hurricane surge, J. Phys. Oceanogr., 38 (9), 2003–2013, https://doi.org/10.1175/2008JPO3727.1, 2008.

Lagmay, A. M. F, Agaton, R. P., Bahala, M. A. C., Briones, J. B. L. T., Cabacaba, K. M. C., Caro, C. V. C., Dasallas, L. L., Gonzalo, L. A. L., Ladiero, C. N., Lapidez, J. P., Mungcal, M. T. F., Puno, J. V. R., Ramos, M. M. A. C., Santiago, J., Suarez, J. K., and Tablazon, J. P.: Devastating storm surges of Typhoon Haiyan, Int. J. Disast. Risk Re., 11, 1–12, https://doi.org/10.1016/j.ijdrr.2014.10.006, 2015.

Rosenzweig, C., and Solecki, W.: Hurricane Sandy and adaptation pathways in New York: Lessons from a first-responder city,  Glob. Environ. Change, 28, 395–408, https://doi.org/10.1016/j.gloenvcha.2014.05.003, 2014.

Xian, S., Feng, K., Lin, N., Marsooli, R., Chavas, D., Chen, J., and Hatzikyriakou, A.: Brief communication: Rapid assessment of damaged residential buildings in the Florida Keys after Hurricane Irma, Nat. Hazards Earth Syst. Sci., 18, 2041–2045, https://doi.org/10.5194/nhess-18-2041-2018, 2018.

Yi, C. J., Suppasri, A., Kure, S., Bricker, J. D., Mas, E., Quimpo, M., and Yasuda, M.: Storm surge mapping of typhoon Haiyan and its impact in Tanauan, Leyte, Philippines, Int. J. Disast. Risk Re., 13, 207–214, https://doi.org/10.1016/j.ijdrr.2015.05.007, 2015.

L40: Change to "governments/local authorities managing coastal areas"

> Changed to "governments/local authorities managing coastal areas" on L40.

L41: Change to "The occurrence of marine natural hazards depends not only on the hazards intensities but also on urban exposure and vulnerability".

> Changed to "The occurrence of marine natural hazards depends not only on the hazards intensities but also on urban exposure and vulnerability".

L45: Omit "created"

> The word "created" was omitted.

L47: Omit "the risk of". Disaster can be initiated by an event, risk is result of the disaster.

> "the risk of" was omitted from L47.

L48: Change "propagate into" to "result in"

> Changed from "propagate into" to "result in" in the manuscript.

L49: Add "In this sense, vulnerability has become..."

> Added "In this sense, vulnerability has become…" to L49 in the manuscript.

L52-53: vulnerability to

> Changed from "vulnerability of" to "vulnerability to".

L53: Add "reducing the consequences of this type..."

> Added "reducing the consequences of this type..." to the manuscript.

L54: Omit "definition and"

> "definition and" was omitted from the manuscript.

L58: "views about"

> Changed from "views of" to "views about" in the manuscript.

L59-61: Rephrase as follows "Based on the theory of sustainable development and from of disaster economics perspective, vulnerability of a system is identified by its ability to prevent and resist a disaster (Turner et al., 2003b)"

> Rephrased to "Based on the theory of sustainable development and from a disaster economics perspective, vulnerability of a system is identified by its ability to prevent and resist a disaster (Turner et al., 2003b) ,"

L65: Change to "Existing studies divide vulnerability into"

> Changed to "Existing studies divide vulnerability into"

L74: What do you mean by "overall place vulnerability"?

> "Overall place vulnerability" means vulnerability covering the whole study area, it also can be called "the whole vulnerability".

L75 Add "critical" before "infrastructure"

> The word "critical" was added before "infrastructure"

L79: Change to "Before 1990s,...was paid...were carried out..."

> Changed to "Before 1990s,...was paid...were carried out..." on L79.

L81: Change to "However, large losses of life and property resulting from the occurrence of more devastating disasters have brought up the attention on the role of social vulnerability in disaster impact."

> Changed to "However, large losses of life and property resulting from the occurrence of more devastating disasters have brought up the attention on the role of social vulnerability in disaster impact."

L86: Change "management" to "assessment". Management involves not only evaluation.

> Changed "management" to "assessment" in L86.

L86-87: Change to "Hence, governments should analyse...policies such as...to improve its adaptation capacity..."

> Changed to "Hence, governments should analyse...policies such as...to improve its adaptation capacity..."

L88: Change to "considerable amount of research...studies on..."

> Changed to "considerable amount of research...studies on..."

L90-91: Change to "Analysis of SV to storm surges...is important due to four main reasons"

> Changed to "Analysis of SV to storm surges...is important due to four main reasons"

L91 "Firstly,...few assessments of...in which...is considered"

> Changed to "Firstly,...few assessments of...in which...is considered" in the manuscript.

L92: Choose between "detailed" and "comprehensive". They are synonyms.

> Removed "detailed and" and kept "comprehensive" in L92.

L92: What is the object of screening? Could you explain the mechanism of screening?

> First, we consider the continuity and availability of data. Second, we must retain the indicators related to the losses caused by storm surges, such as fishery output value and port cargo throughput. Third, we reserve the indicators that reflect the strength of disaster prevention and mitigation, such as regional GDP. Finally, we retain the weak indicators at the time of the disaster, such as female proportion and total enrollment of students.

L92: The process of screening itself cannot be a buffer against disaster.

> Reply with above paragraph.

L94: Change to "other coastal cities, which are exposed to similar or other types of marine natural hazards." Once more, authors need to specify if and how the methodology was tailored to hazards resulting from storm surges. As it is, it is applicable to any disaster causing floods (and not only), which means the authors should reconsider the title or give additional emphasis on the hazard intensities and extents related to storm surges; in other words, to further justify their choice of disaster.

> Changed to "other coastal cities, which are exposed to similar or other types of marine natural hazards." in the manuscript. Storm surge is the one of the most dangerous natural disasters that happen in Shenzhen. As stated above, we explain the mechanism of screening dedicators. We will test and validate the indicator system using loss data for storm surges, contained in the *China National Marine Disaster Bulletin*.

L95: Change to "Secondly, since 1979, political reform and openness has led...in Shenzhen." Omit "during the study period"

> Changed to "Secondly, since 1979, political reform and openness has led...in Shenzhen." and omitted "during the study period" from L95.

L95: Omit "expedited process"; "rapid" or "accelerated" is enough to describe the process.

> "expedited process" was omitted from L95 and "rapid" was kept to describe the process.

L97: Change to "By choosing Shenzhen, we study a typical scenario of SV change as a result of..."

> Changed to "By choosing Shenzhen, we study a typical scenario of SV change as a result of..."

L98: "Thirdly, so far,..."

> Changed to "Thirdly, so far,..." in the updated manuscript.

L100: "Instead, herewith, a composite…"

> Changed to "Instead, herewith, a composite…" in the manuscript.

L102: "Data envelopment analysis (DEA)...evaluation in China to discover..."

> Changed to "Data envelopment analysis (DEA)...evaluation in China to discover..." in the manuscript.

L105-106: Rephrase "Five methods for combined evaluation were used by Liu and Liu (2017) and results determined that among seven coastal cities selected for evaluation in Shandong Province Yantai city and Binzhou city had the highest and lowest vulnerability, respectively."

> Rephrased to "Five methods for combined evaluation were used by Liu and Liu (2017). Their results determined that among seven coastal cities in Shandong Province selected for evaluation, Yantai city and Binzhou city had the highest and lowest vulnerability, respectively."

L108: Rephrase "The socioeconomic vulnerability to typhoon-induced storm surges for municipal districts of Guangdong Province was assessed using the fuzzy comprehensive evaluation. It was determined that vulnerability presented a large spatial heterogeneity (Zhang et al., 2010)."

> Rephrased to "The socioeconomic vulnerability to typhoon-induced storm surges for municipal districts of Guangdong Province was assessed using the fuzzy comprehensive evaluation. It was determined that vulnerability presented a large spatial heterogeneity (Zhang et al., 2010)."

L111: Omit "with results from Zhang et al. (2010)"

> "with results from Zhang et al. (2010)" was omitted from L111.

L112: Change "differences" to "dimensions".

> Changed from "differences" to "dimensions".

L115: Temporal patterns = trend ?

> Changed from "Temporal patterns" to "trends" in the manuscript.

L116: Can you explain what you mean by "macroscopic angle"? Here, you can state again the period your research covers.

> The macroscopic angle means that the scale of the evaluation is based on the whole city but not street or community scales.

L122: Change to "Since its establishment in 1979, in just 40 years,...through a..."

> Changed to "Since its establishment in 1979, in just 40 years,...through a..." in the manuscript.

L124: Change to "However, due to its location at the coast of the Pearl River Delta (Fig. 1a,b) and its proximity to the northern part of the South China Sea (Fig. 1b,c), Shenzhen is facing many coastal disasters threatening its sustainable development, among which storm surge induced disasters are the most severe."

> Changed to "However, due to its location at the coast of the Pearl River Delta (Fig. 1a,b) and its proximity to the northern part of the South China Sea (Fig. 1b,c), Shenzhen is facing many coastal disasters threatening its sustainable development, among which storm surge induced disasters are the most severe."

L127:"[http://www.sz.gov.cn/ytqzfzx/yingji/yjya/201712/t20171206_10111758.htm (last access: 30 June 2019)]" This should be moved to the list of references.

> This direct URL was already present in the following citation, located in the references section, and was removed from L127:

Shenzhen Marine Disaster Emergency Plan, Retrieved from http://www.sz.gov.cn/ytqzfzx/yingji/yjya/201712/t20171206_10111758.htm (last access: 30 June 2019), 2017

L129: "116 typhoons have seriously affected the Shenzhen coastal area"

> Changed to "116 typhoons have seriously affected the Shenzhen coastal area" in the manuscript.

L136: Maybe you mean "The increased frequency of storm surges"

> Changed to "The increased frequency of storm surges" in the manuscript.

L137-138: It is not clear if EWS is a recommendation? Can you relate this to your research results? Which are those "particularly susceptible areas"?

> It is valuable to assess the social vulnerability to storms surges for Shenzhen in order to provide necessary support for the government to improve the level of disaster prevention.

L140: Change "fully contained" to "entirely available"

> Changed from "fully contained" to "entirely available" in the manuscript.

L141-142: Change "which was compiled by the Shenzhen Statistical Bureau and a Shenzhen-based investigation team of the National Bureau of Statistics, and published (updated annually) by the Shenzhen Statistical Bureau." to "which is compiled and published on annual basis".

> Changed from "which was compiled by the Shenzhen Statistical Bureau and a Shenzhen-based investigation team of the National Bureau of Statistics, and published (updated annually) by the Shenzhen Statistical Bureau." to "which is compiled and published on annual basis".

L153-154: Change to "Due to the absence of long-term statistical data on some important indicators, this study is limited to a partial statistical dataset spanning the period 1986 - 2016 in order to sustain the data integrity."

> Changed to "Due to the absence of long-term statistical data on some important indicators, this study is limited to a partial statistical dataset spanning the period 1986–2016 in order to sustain the data integrity."

L165: By "evaluation result", don't you mean "final score"?

> Yes, we mean 'final score'.

L175-178: Provide citations for all mentioned methods and tests.

> Thanks for your suggestion, as citations have been added to the updated manuscript.

L179&201: Method or strategy?

> Strategy has been deleted. The correct expression is "combined weighting method"

L179-181: Rephrase as follows "Finally, the combined evaluation results are achieved,which have significant advantages compared to those of all single methods due to weighted value of each evaluation method."

> Rephrase as follows "Finally, the combined evaluation results are achieved, which have significant advantages compared to those of all single methods due to weighted value of each evaluation method."

L183-184: It is not clear if the first sentence describe a statement or research action.

> The first sentence is a research action.

L187: I cannot understand what the meaning of "knowledge simplicity attribute of rough set" is.

> It means that rough set theory can simplify knowledge and extract the main information from it.

L188: Among which?

> Deleted "among them"

L199: Change "all above evaluation methods" to "all evaluation methods in use"

> Changed from "all above evaluation methods" to "all evaluation methods in use".

L201: Using "a single evaluation framework", do you refer to combined weighting method?

> Yes, we refer to the combined weighting method.

L213: Change to "Calculate the proportion of the indicator j in year i (rij).

> Changed to "Calculate the proportion of the indicator j in year i (rij)." in the manuscript.

L234: The meaning of the second sentence in the paragraph is not clear.

> We will simplify and clarify the introduction of these five methods in one section as you recommend.

L260: Due to limitations of the methods in use, each single evaluation can lead to a different conclusion."

> We will organize this section again.

L260: Use "Nevertheless" instead of "However". Section 2.4 is really hard to follow. Could you better explain the connection between methods (section 2.3) and resulting indicators (section 2.4)?

> We used "Nevertheless" instead of "However". We have adjusted the content in those sections of the paper.

L286-288: It is not clear if this is achieved within previous researches or is a stage within the present study.

> It is achieved with the present study with the previous researches as reference.

L289: Storm surges are caused by the action of tropical and extratropical cyclones not accompanied by them.

> Changed "accompanied" to "caused" in L289.

L290: Could you explain better the screening process? What is the reason to take out of consideration man-made barriers?

> Explained above in L92. Urban fixed asset investments may reflect the consideration of man-made barriers.

L291: To what disaster body do you refer: the city itself or a body in general? Which are the bodies you screen?

> Body is not used here accurately. We want to say that different aspects are influenced by the disaster. For example, storm surge can destroy roadways, injure people, disable transportation, etc. We will correct the expression in this section.

L291-293: The sentences seem to repeat one another. Could you clarify and rephrase.

> Delete "As for the exposure of a disaster body, this research selects key indicators that are highly accessible and can reflect a disaster-stricken area at a macro level."

L304: Change "While regional GDP" to "Since the amount of regional GDP"

> Changed from "While regional GDP" to "Since the amount of regional GDP".

L305: Change "equates" to "corresponds", "for" to "to". The last two sentences in section 2.4.1 should change their places.

> Changed "equates" to "corresponds" and changed "for" to "to". The last two sentences in **Section 2.4.1** were switched.

L312: Again, is Shenzhen city the disaster body? Industries of primary importance for Shenzhen city are…

> This was explained above for L291.

L313: Change "fluctuations" to "changes"

> Changed "fluctuations" to "changes" in the updated manuscript.

L315: Change "higher winds and precipitation patterns" to "severe winds and precipitations"

> Changed "higher winds and precipitation patterns" to "severe winds and precipitations"

L316: Change "inconvenient" to "busy"

> Changed "inconvenient" to "busy" in the manuscript.

L317: Change "suffer casualties outside" to "suffer injuries or even cause casualties"

> Changed from "suffer casualties outside" to "suffer injuries or even cause casualties".

L322: Change "with which" to "meaning that"

> Changed from "with which" to "meaning that".

L324: Change "aspects" to "groups"

> Changed "aspects" to "groups" in the updated manuscript.

L326: Change "more money is devoted into" to "more resources are provided/spent for"

> Changed "more money is devoted into" to "more resources are provided/spent for".

L330: Add "consequences" after "resists disaster"

> Added "consequences" after "resists disaster".

L331: "per capita"

> Changed to "per capita" in the updated manuscript.

L333: Change "infrastructure construction" to "public services"

> Changed from "infrastructure construction" to "public services" in the updated manuscript.

L334-335: Change to "level of medical and health care, including the number of medical and health institutions and their equipment (e.g. beds etc...) as well as the number of health employees."

> Changed to "level of medical and health care, including the number of medical and health institutions and their equipment (e.g. beds etc...) as well as the number of health employees."

L336: "potential victims"

> Changed to "potential victims" in the updated manuscript.

L342: Change "social vulnerability to storm surges discussed in this research can be approximately divided into" to "degrees of social vulnerability to storm surges discussed in this research are set to..."

> Change "social vulnerability to storm surges discussed in this research can be approximately divided into" to "degrees of social vulnerability to storm surges discussed in this research are set to..."

L345: It is not clear if proposed SVI threshold values were calculated by the authors or were borrowed from Yuan et al. (2016).

> We changed the standard and calculated the threshold values by ourselves.

L346: Change "close" to "similar"

> Changed "close" to "similar" in the updated manuscript.

L422: Change "obvious" to "pronounced", "variation" – "variability"

> Changed "obvious" to "pronounced" and "variation" to "variability" in the manuscript.

Figure 2 is not referred to in the text.

> We will refer to Figure 2 in the first paragraph of **Section 2.1** and explain it.

L753: Change to "and outlined using crimson colour."

> Changed to "and outlined using crimson colour." in the updated manuscript.

L836: Change to "method (blue line).  The weighted value of SVI is depicted with thick red line."

> Changed to "method (blue line).  The weighted value of SVI is depicted with a thick red line." in the updated manuscript.

Table1/3: Resilience: Per capita

> Changed from "Resilience" to "Resilience: Per capita" in both Table 1 and Table 3.

Acknowledgements: Modification

> Modified the Acknowledgements section to reflect the changes in work ownership and figure descriptions.

Figure 2: Replacement

> Replaced original Figure 2 with a Python version Figure 2, for consistency among all other plots.

---

## Author Response (AR1)

**Author's Response to Anonymous Reviewer #1**

The research group thanks Anonymous Reviewer #1 for their detailed comments and supportive suggestions. This feedback allows our group to make appropriate updates to the manuscript in preparation for uploading an improved version.

An author's response to Anonymous Reviewer #1 is denoted with a > symbol and amber text. Some confirmed additions are placed in magenta text and are located at the end of the **Author's Response to Anonymous Reviewer #2**.

General comments:

(1) Although the impact of storm surge has been discussed in the introduction, there is no indication of how this study is specifically linked to storm surge. All analyses in this study appear to be based on data of the entire Shenzhen rather than the coastal regions of Shenzhen which are actually vulnerable to storm surge risk. This study appears to be an attempt to quantify the social vulnerability of Shenzhen to all types of natural disaster. Thus the authors might want to reconsider the title of this study.

> Yes, it is a good observation about this paper. Due to a data acquisition limitation, it is very difficult or impossible to find perfect indicators with a long-term record specifically linked to storm surge in China. This work is a creative attempt to analyze publicly available "macroscopic" data in order to explain the "microscopic" phenomena for such similar Chinese coastal cities. Furthermore, the data's spatial coverage is a narrow, 20 km length in the north-south direction across Shenzhen City and therefore most areas of Shenzhen are threatened during storm surges. Additionally, some directly related indicators of storm surge are selected, rather than other types of natural disasters such as "fishery output value" and "port cargo throughput", and used in the indicator system for evaluating social vulnerability. We believe our results address the problem to a certain extent. From this research, it becomes feasible for us to deliver suggestions to local governments about the need to collect and archive statistical data for most threatened coastal communities. We made an appropriate title change to "Trends in social vulnerability to storm surges in Shenzhen, China".

(2) In this study, the authors did not establish any connection between their social vulnerability index (SVI) and storm surges, i.e. validation of the SVI is not included. For example, Su et al. (2015) used total economic loss of hazards to examine the performance of their SVI which consequently show their SVI is linked to loss due to hazards. The authors could address this problem by relating SVI to economic loss, number of injuries due to storm surges, number of fatalities due to storm surges etc.

> Yes, this is a very constructive idea. We considered adding a validation strategy but public data of economic loss due to storm surges in Shenzhen is simply unavailable. Economic loss data was found to be available for Guangdong province (i.e. a broader scale dataset) which is where Shenzhen is located. We assume that the loss from a storm surge disaster in Shenzhen and Guangdong province are in direct proportion. We added a validation section to this manuscript. It is located in **3.3 Validation of SVI to storm surges**.

(3) There are a lot of statements and claims in this paper but the authors did not include the source of information or the studies to support those statements and claims. (Some of them are listed in the specific comments).

> Yes, thank you for your reminder. We added all necessary sources of information and additional citations to support our statements and claims.

Specific comments (Technical corrections):

Page 2, lines 36-38: Reference is needed.

> These citations were added to the manuscript:

Forbes, C., Rhome, J., Mattocks, C., and Taylor, A. A.: Predicting the storm surge threat of Hurricane Sandy with the National Weather Service SLOSH model, J. Mar. Sci. Eng., 2 (2), 437–476, https://doi.org/10.3390/jmse2020437, 2014.

Frank, N. L., and Husain, S. A.: The deadliest cyclone in history? Bull. Am. Meteor. Soc., 52 (6), 438–445, 1971.

Fritz, H. M., Blount, C., Sokoloski, R., Singleton, J., Fuggle, A., McAdoo, B. G., Moore, A., Grass, C., and Tate, B.: Hurricane Katrina storm surge distribution and field observations on the Mississippi Barrier Islands, Estuar. Coast. Shelf Sci., 74 (1-2), 12–20, https://doi.org/10.1016/j.ecss.2007.03.015, 2007.

Fritz, H. M., Blount, C., Thwin, S., Thu, M. K., and Chan, N.: Cyclone Nargis storm surge in Myanmar, Nature Geosci., 2 (7), 448–449, https://doi.org/10.1038/ngeo558, 2009.

Irish, J. L., Resio, D. T., and Ratcliff, J. J.: The influence of storm size on hurricane surge, J. Phys. Oceanogr., 38 (9), 2003–2013, https://doi.org/10.1175/2008JPO3727.1, 2008.

Lagmay, A. M. F, Agaton, R. P., Bahala, M. A. C., Briones, J. B. L. T., Cabacaba, K. M. C., Caro, C. V. C., Dasallas, L. L., Gonzalo, L. A. L., Ladiero, C. N., Lapidez, J. P., Mungcal, M. T. F., Puno, J. V. R., Ramos, M. M. A. C., Santiago, J., Suarez, J. K., and Tablazon, J. P.: Devastating storm surges of Typhoon Haiyan, Int. J. Disast. Risk Re., 11, 1–12, https://doi.org/10.1016/j.ijdrr.2014.10.006, 2015.

Rosenzweig, C., and Solecki, W.: Hurricane Sandy and adaptation pathways in New York: Lessons from a first-responder city, Glob. Environ. Change, 28, 395–408, https://doi.org/10.1016/j.gloenvcha.2014.05.003, 2014.

Xian, S., Feng, K., Lin, N., Marsooli, R., Chavas, D., Chen, J., and Hatzikyriakou, A.: Brief communication: Rapid assessment of damaged residential buildings in the Florida Keys after Hurricane Irma, Nat. Hazards Earth Syst. Sci., 18, 2041–2045, https://doi.org/10.5194/nhess-18-2041-2018, 2018.

Yi, C. J., Suppasri, A., Kure, S., Bricker, J. D., Mas, E., Quimpo, M., and Yasuda, M.: Storm surge mapping of typhoon Haiyan and its impact in Tanauan, Leyte, Philippines, Int. J. Disast. Risk Re., 13, 207–214, https://doi.org/10.1016/j.ijdrr.2015.05.007, 2015.

Page 4, lines 121-122: Reference is needed.

> We used the following citation for line 121-122:

Zünd, D. and Bettencourt, L. M. A.: Growth and development in prefecture-level cities in China, PLoS ONE, 14(9), e0221017, https://doi.org/10.1371/journal.pone.0221017, 2019.

> Also, we changed from "attributed to the highest per capita Gross Domestic Product (GDP) in mainland China" to "attributed to one of the highest Gross Domestic Product (GDP) per capita in mainland China...".

Page 6, lines 168-169: Full form of AHP and PCA are needed.

> The terms analytic hierarchy process and principal component analysis were added to the text.

Page 6, lines 175-176, 178: References for these methods would be needed.

> We replaced the words with "Firstly, the construction of an optimized social vulnerability evaluation indicator system, based on the idea of rough set theory (Das et al., 2018), is completed.  Second, the entropy method (Zhou and Yang, 2019), the Technique for Order Preference by Similarity to an Ideal Solution (TOPSIS) method (Kuo, 2017) and the coefficient of variation method (Zhou et al., 2004) are used to weigh the indicators and aggregate SVI separately.  Then, the consistency of different evaluation results is tested by using the compatibility test method, i.e., Kendall consistency test (Wen and Hu, 2002)."  All of the references have been included already.

Page 6, lines 183-184: What is "a theoretical framework" referring to?

> Changed "a theoretical framework" to "vulnerability theory"

Page 7, line 216, equation 2: What does "lnn" mean?

> Equation (2) was changed to:

$$e_j = -(\ln n)^{-1} \sum_{i=1}^{n} \bar{r}_{ij} \ln \bar{r}_{ij} \ (0 \le e_j \le 1, j = 1,2,3,\cdots,m)$$

Page 10, line 289-290: I doubt that Shenzhen faces storm surges accompanied by extratropical cyclones on regular basis. Could you please provide studies to support your statement?

> We deleted "extratropical cyclones" from the manuscript, i.e. line 289.

Page 11, lines 314-319: I am not sure about your argument here.  Could you please provide studies/evidence that support your claim "students at school and women are more likely to suffer casualties outside" due to the harsh meteorological conditions?  Why elderly people and people with disability are not included in the vulnerable groups?  Could you please provide studies/evidence that support your claim about "social workers"?  Are the authors assuming (all) people would still go to school/work despite a typhoon is affecting the city? As far as I know, when certain typhoon warning signal (yellow, orange, and red) is issued, school and work would be suspended.  People would be asked to stay inside a safe building or evacuate to a safe location, i.e. majority of the people should be in safe locations. Thus I am not sure why students at school, women, and social workers are explicitly included as sensitivity indicators.

> Thanks for your detailed observation and relevant questions. With regard to the sensitivity indicators, as we know, the occurrence of storm surges is uncertain and the early warning system is not that accurate. Unfortunately, when a strong storm surge occurrence happens, it is difficult to have all students, women and social workers be held in the safe place. Additionally, elderly people and people with disabilities are included in vulnerable groups (Yuan et al., 2016), but there is no specific data captured about the elderly population in Shenzhen's statistical yearbooks. We would like to include all factors that would reach the general agreement of the marine disaster community. However, due to the lack of original data, we can only provide certain factors to the indicator system for analysis.

> We will add the following citation to line 317:

Yuan, S., Zhao, X., and Li, L. L.: Combination evaluation and case analysis of vulnerability of storm surge in coastal provinces of China, Haiyang Xuebao., 38 (2), 16–24, https://doi.org/10.3969/j.issn.0253-4193.2016.02.002, (in Chinese), 2016.

Page 12, lines 331-333: Please provide evidence to support your claim – high income level of residents and higher living standard implies strong disaster resilience and faster post-disaster recovery.

> We added the following citation to line 333:

Yuan, S., Zhao, X., and Li, L. L.: Combination evaluation and case analysis of vulnerability of storm surge in coastal provinces of China, Haiyang Xuebao., 38 (2), 16–24, https://doi.org/10.3969/j.issn.0253-4193.2016.02.002, (in Chinese), 2016.

Page 12, lines 342-345: It is not clear how does the categorisation of the index, which is developed by Yuan et al. (2016), can be applied to the SVI, which is developed in the current study. These 2 indices do not have the same composition! In addition, the interpretation of this categorisation is not clear. How should we use this categorisation?

> We adjusted the categorisation in the updated manuscript.

Regarding lines 342-345:

Changed sentence "According to previous studies on disaster vulnerability, social vulnerability to storm surges discussed in this research can be approximately divided into (i) high vulnerability, (ii) relatively high vulnerability, (iii) moderate vulnerability, (iv) relatively low vulnerability and (v) low vulnerability and the corresponding critical points of SVI are 0.5873, 0.5163, 0.4452 and 0.3741, respectively (Yuan et al., 2016)."
to
"According to the common idea of equal division in mathematical statistics, social vulnerability to storm surges discussed in this research can be approximately divided into (i) high vulnerability, (ii) relatively high vulnerability, (iii) moderate vulnerability, (iv) relatively low vulnerability and (v) low vulnerability and the corresponding critical points of SVI are 0.5715, 0.5237, 0.4759 and 0.4281, respectively."

Regarding lines 355-358:

Changed sentence "According to classification criteria, social vulnerability to storm surges in Shenzhen during the entire study period can be divided into four stages: (i) high social vulnerability between

1986 to 1992, (ii) relatively high social vulnerability between 1993 to 2008, (iii) moderate social vulnerability between 2009 and 2014, and (iv) relatively low social vulnerability between 2015 and 2016. The time to maintain relatively high (low) social vulnerability is the longest (shortest) as a whole, respectively."
to
"According to classification criteria, social vulnerability to storm surges in Shenzhen during the entire study period can be divided into five stages: (i) high social vulnerability between 1986 to 1994 and 1999 to 2004, (ii) relatively high social vulnerability between 1995 to 1998 and 2005 to 2008, (iii) moderate social vulnerability between 2009 to 2013, (iv) relatively low social vulnerability in 2014 and (v) low vulnerability in 2015 and 2016. The time to maintain high social vulnerability is the longest and relatively low social vulnerability is the shortest as a whole, respectively. It is apparent that, after 2008, social vulnerability has been completely removed from relatively high levels."

Page 13, line 370, 371: Please provide the full form of EI, SI, and RI in the main text.

> The terms exposure index, sensitivity index and resilience index were added to the text. - Completed

Page 13, line 374-375: Reference is needed as this information cannot be found in Figure 5.

> We understand the reviewer's concern. Fiscal spending, residents' income levels, completion degree of medical conditions, and infrastructure are all included in the indicator system and they all belong to resilience indicators. In Figure 5, we can see the continuous decrease of RI and that is caused by the improvement of Shenzhen's fiscal spending, residents' income levels, completion degree of medical conditions, and infrastructure. While the conclusion cannot be found in Figure 5, it can be indicated by the trend of RI. Therefore, we believe a reference is not needed here.

Figure 2: Figure 2 is not mentioned in the main text! Please name the source(s) of the GDP data set.

> We added "Through the growth of GDP, it is found that Shenzhen's economic level is progressively advancing during our study period (Fig. 2)." at the end of the first paragraph in Section **2.1 Study area and data sources**. The source of the GDP data set is the same as Section **2.1 Study area and data sources**.

Figure 5: It is clear that the downward trend of SVI is mainly driven by the downward trend of RI. However, it is not clear that whether RI could be negative. If RI can be negative, how should it be interpreted? If RI cannot be negative, does this study suggest there exists a threshold in RI that social vulnerability cannot be reduced by further improvement of city's resilience?

> It's true that the downward trend of SVI is mainly driven by the downward trend of RI, but RI cannot be negative. In 2016, all of the resilience indicators happen to reach their maximum, and when performing normalization, the resilience index in 2016 equals zero (zero is the minimum of RI). Social vulnerability can be reduced by further improvement of the city's resilience because when RI equals zero, it only indicates that the resilience for that year is the strongest compared with other years during the whole study period, rather than the city improving its resilience to the largest degree.

Figures 6, 7: I am not sure what does "normalized values" mean.

> The graph objects are all normalized to a uniform range for comparison. Since their dimensions are different, they are uniformly converted into dimensionless quantities. In figure 6, we use min-max

normalization which means a linear transformation is performed on the original data so that the result falls into the interval [0,1]. In figure 7, we also use min-max normalization but the value of indicators fall into the interval [0,0.25]. For SVI, we subtract 0.38 from the value to yield an interval [0,0.25]. As a result, we can easily compare the relationship between variables.

**Author's Response to Anonymous Reviewer #2**

The research group thanks Anonymous Reviewer #2 for their careful review, constructive feedback and technical screening. We know the reviewer's comments, suggestions and list of technical corrections will allow our group to forge an enhanced version of the manuscript.

An author's response to Anonymous Reviewer #2 is denoted with a > symbol and amber text. Some confirmed additions are placed in magenta text and are located at the end of this response.

**General comments:**

The proposed research presents an effort to assess the long-term trend in social vulnerability to storm surge induced flood hazard in Shenzhen, China using a system of SV indicators. It was constructed using a complex approach consisting of combination and weighting of results obtained through application of three single evaluation methods. The work is interesting in the context of preparedness, mitigation and adaptation of a large city to natural disaster impacts. The study is well situated among the existing regional assessments and has well defined scope. Authors show fluency in applying and interpreting the risk theory. The number and relevance of proposed indicators are suitable. The assessment of contribution of indicators to final SVI is particularly valuable. Conclusion are very well structured. Such a study is certainly useful for wide range of stakeholders, especially policy makers, local authorities and coastal managers. Nevertheless, there are several issues that need to be addressed. The research group is thankful for your professional assessment and honest feedback. The group is pleased to hear that the research will be useful to a large range of stakeholders and scientific communities.

**Specific comments:**

(1) As authors themselves pointed out the proposed approach can be used to assess social vulnerability for a variety of disasters, of which they had chosen storm surges. However, detailed enough information on the storm surge induced flood hazard intensity and extent is not provided as well as which coastal areas are most susceptible.

> Yes, detailed information on storm surge-induced flood hazard intensity and extent is provided. Information about coastal areas, which are most susceptible, is located in a new second paragraph of Section **1. Introduction**. Also, we've provided detailed information about storm surge induced flood hazard intensity and how it relates to loss data, located in **3.3 Validation of SVI to storm surges**.

(2) Evidently, the lowest level of regional disaggregation is the city of Shenzhen and SVI is relevant for the entire city but not for separate districts within the city limits. This could be a problem with the selected hazard as there is no distinction between areas,which are under direct and indirect hazard impact. Although this is not crucial for a SVI, a better distinction could raise the value of the proposed research. As it is, the study provides only a broad view of the SV in the region. I wonder if there is a possibility to focus on the communities, which are most threatened by flood hazard.

> Yes, we understand your concern. We'd like to focus on small scale districts and communities which are most threatened by storm surge but it is difficult or even impossible to obtain community data at this spatial resolution. This work is a creative attempt to analyze publicly available "macroscopic" data in order to explain the "microscopic" phenomena for such similar Chinese coastal cities. Furthermore, the data's spatial coverage is a narrow, 20 km length in the north-south direction across Shenzhen City

and therefore most areas of Shenzhen are threatened during storm surges. We believe our results address the problem to a certain extent. From this research, it becomes feasible for us to deliver suggestions to local governments about the need to collect and archive statistical data for most threatened coastal communities.

(3) I would change "temporal variability to" to "trends in" in the title since a process (variable) can (significantly) vary temporally while SV (despite of minor fluctuations) is more likely to exhibit increasing or decreasing trend over a certain period of time (as discovered by the authors).

> Yes, it is a very good suggestion. We made a title change from "temporal variability to" to "trends in" in the updated manuscript. The full title now reads "Trends in social vulnerability to storm surges in Shenzhen, China".

(4) Compared to the Method chapter, Results & Discussion one seems a bit under developed. I think readers would appreciate a more in depth exposition. The paper's chapters are overly subdivided, which makes the text somewhat choppy. It is suggested to decrease the number of subdivisions.

> Yes, you are correct and we agree with you. We condensed the sections in **2. Materials and methods**. We reduced subsections under **2.3 Indicator system of social vulnerability evaluation**. The **3. Results and discussion** section were balanced (content wise) with the other chapters, so research results are expressed in a more holistic manner. The current sections are now presented as:

1. Introduction
2. Materials and methods
 2.1 Study area and data sources
 2.2 Research methods
  2.2.1 Entropy method
  2.2.2 TOPSIS method
  2.2.3 Coefficient of variation method
  2.2.4 Kendall consistency test
  2.2.5 Combination weighting method
 2.3 Indicator system of social vulnerability evaluation
3. Results and discussion
 3.1 Variation characteristics of social vulnerability
 3.2 Reasons for vulnerability changes
 3.3 Validation of SVI to storm surges
4. Conclusion

(5) Although the English language used throughout the manuscript is generally correct, it would still need grammar improvements. The text is sometimes rather heavy to follow and understand so stylistic upgrade is also recommendable.

> Yes, we do agree with you. In the updated manuscript, the grammar and English language was improved. We worked on reducing the heaviness of the text and we balanced the content better especially in Sections **2.x**, **3.x** for readability purposes.

**Technical corrections:**

A detail list of advisable corrections and specific questions follows. It is solely meant to increase the paper impact by improving the grammar, text fluency and better understanding of the complex connections between methods, resulting indicators and study findings:

The research group thanks Anonymous Reviewer #2 for providing us this detailed list of technical corrections and suggestions.

L16: Change to "Evaluation of social vulnerability to storm surges is important for any coastal city to provide..."

> We changed to "An evaluation of social vulnerability to storm surges is important for any coastal city to provide..."

L19: Change to "which are subsequently combined by weighting in order to calculate a common SVI."

> We changed to "which are subsequently combined by weighting in order to calculate a common social vulnerability index."

L21-22: Split into 2 sentences

> These sentences now read as "Shenzhen has a current reputation of having the most economic development potential and is a representative city in China.  The city is chosen to evaluate its social vulnerability to storm surges via a historical social and economic statistical dataset spanning from 1986–2016." in the updated manuscript.

L22: "The research extends further by analysing the city's temporal variability." This sentence is not clear. Temporal variability of what?

> The sentence "The research extends further by analyzing the city's temporal variability." was deleted.

L25: Change to "continuous increase of medical services supply"

> We changed to "...continuous increase of medical services supply..."

L26-27: Exposure and sensitivity define vulnerability hence should precede it in the exposition.

> We changed from "Results reveal that social vulnerability keeps almost constant from 1986–1991 and 1993–2004, while it decreased sharply in the remainder of times to show a 'stair-type' declining curve over the past 30 years.  Resilience is progressively increasing by virtue of a continuous increase in medical institutions, fixed asset investments and salary levels of employees.  These determinants contribute to the overall downward trend of social vulnerability for Shenzhen.  Exposure and sensitivity increased slowly with some fluctuation, causing the changes of social responsibility to transpire."
to
"Exposure and sensitivity increased slowly with some fluctuation, leading to fluctuations in the trends of social vulnerability. Social vulnerability keeps almost constant during 1986–1991 and 1993–2004, while it decreased sharply in the rest of the time to form a 'stair-type' declining curve over the past 30 years. Resilience is progressively increasing by virtue of a continuous increase of medical services

supply, fixed asset investments and salary levels of employees. These determinants contribute to the overall downward trend of social vulnerability for Shenzhen."

L23-24: Change to "during 1986–1991 and 1993–2004… in the rest of the time to form..."

> We changed to "during 1986–1991 and 1993–2004… in the rest of the time to form..."

L27: What do you mean by "causing the changes of social responsibility to transpire"?

> We changed from "causing the changes of social vulnerability to transpire." to "leading to fluctuations in the trends of social vulnerability."

Keywords preferably should not repeat the title. Only "Combined evaluation; Indicator system" give additional insight to the study.

> Yes, it is a good suggestion. We used the following keywords: "Social vulnerability; Storm surge; Indicator system; Shenzhen, China" in the updated manuscript.

L32: Change to "during transition of"

> We changed to "during transition of" in the updated manuscript.

L32-34 Split the first sentence into two sentences

> The sentence was split into two sentences and reads as "Storm surge refers to the abnormal volumetric rise of sea water layered above the astronomical tide due to severe meteorological conditions experienced during transition of low-pressure weather systems.  Tropical and extratropical cyclones rank near the pinnacle among marine natural hazards in terms of human casualties and expensive infrastructure losses."

L33: Omit "historical counts of"

> We removed "historical counts of" from L33 in the latest manuscript.

L35: Change "ability" to "potential"

> We changed from "ability" to "potential" in the updated manuscript.

L37-38: Provide citations for mentioned catastrophic events

> These citations were added to the updated manuscript:

> Forbes, C., Rhome, J., Mattocks, C., and Taylor, A. A.: Predicting the storm surge threat of Hurricane Sandy with the National Weather Service SLOSH model, J. Mar. Sci. Eng., 2 (2), 437–476, https://doi.org/10.3390/jmse2020437, 2014.

Frank, N. L., and Husain, S. A.: The deadliest cyclone in history? Bull. Am. Meteor. Soc., 52 (6), 438–445, 1971.

Fritz, H. M., Blount, C., Sokoloski, R., Singleton, J., Fuggle, A., McAdoo, B. G., Moore, A., Grass, C., and Tate, B.: Hurricane Katrina storm surge distribution and field observations on the Mississippi Barrier Islands, Estuar. Coast. Shelf Sci., 74 (1-2), 12–20, https://doi.org/10.1016/j.ecss.2007.03.015, 2007.

Fritz, H. M., Blount, C., Thwin, S., Thu, M. K., and Chan, N.: Cyclone Nargis storm surge in Myanmar, Nature Geosci., 2 (7), 448–449, https://doi.org/10.1038/ngeo558, 2009.

Irish, J. L., Resio, D. T., and Ratcliff, J. J.: The influence of storm size on hurricane surge, J. Phys. Oceanogr., 38 (9), 2003–2013, https://doi.org/10.1175/2008JPO3727.1, 2008.

Lagmay, A. M. F, Agaton, R. P., Bahala, M. A. C., Briones, J. B. L. T., Cabacaba, K. M. C., Caro, C. V. C., Dasallas, L. L., Gonzalo, L. A. L., Ladiero, C. N., Lapidez, J. P., Mungcal, M. T. F., Puno, J. V. R., Ramos, M. M. A. C., Santiago, J., Suarez, J. K., and Tablazon, J. P.: Devastating storm surges of Typhoon Haiyan, Int. J. Disast. Risk Re., 11, 1–12, https://doi.org/10.1016/j.ijdrr.2014.10.006, 2015.

Rosenzweig, C., and Solecki, W.: Hurricane Sandy and adaptation pathways in New York: Lessons from a first-responder city, Glob. Environ. Change, 28, 395–408, https://doi.org/10.1016/j.gloenvcha.2014.05.003, 2014.

Xian, S., Feng, K., Lin, N., Marsooli, R., Chavas, D., Chen, J., and Hatzikyriakou, A.: Brief communication: Rapid assessment of damaged residential buildings in the Florida Keys after Hurricane Irma, Nat. Hazards Earth Syst. Sci., 18, 2041–2045, https://doi.org/10.5194/nhess-18-2041-2018, 2018.

Yi, C. J., Suppasri, A., Kure, S., Bricker, J. D., Mas, E., Quimpo, M., and Yasuda, M.: Storm surge mapping of typhoon Haiyan and its impact in Tanauan, Leyte, Philippines, Int. J. Disast. Risk Re., 13, 207–214, https://doi.org/10.1016/j.ijdrr.2015.05.007, 2015.

L40: Change to "governments/local authorities managing coastal areas"

> We changed to "governments/local authorities managing coastal areas" on L40.

L41: Change to "The occurrence of marine natural hazards depends not only on the hazards intensities but also on urban exposure and vulnerability".

> We changed to "The occurrence of marine natural hazards depends not only on the hazards intensities but also on urban exposure and vulnerability".

L45: Omit "created"

> The word "created" was removed in the updated manuscript.

L47: Omit "the risk of". Disaster can be initiated by an event, risk is result of the disaster.

> We removed "the risk of" in the latest manuscript.

L48: Change "propagate into" to "result in"

> We changed from "propagate into" to "result in" in the latest manuscript.

L49: Add "In this sense, vulnerability has become..."

> We added "In this sense, vulnerability has become…" to L49 in the manuscript.

L52-53: vulnerability to

> We changed from "vulnerability of" to "vulnerability to" in the latest manuscript.

L53: Add "reducing the consequences of this type..."

> We added "reducing the consequences of this type..." to the latest manuscript.

L54: Omit "definition and"

> We removed "definition and" in the latest manuscript.

L58: "views about"

> We changed from "views of" to "views about" in the latest manuscript.

L59-61: Rephrase as follows "Based on the theory of sustainable development and from of disaster economics perspective, vulnerability of a system is identified by its ability to prevent and resist a disaster (Turner et al., 2003b)"

> We rephrased to "Based on the theory of sustainable development and from a disaster economics perspective, vulnerability of a system is identified by its ability to prevent and resist a disaster (Turner et al., 2003b)."

L65: Change to "Existing studies divide vulnerability into"

> We changed to "Existing studies divide vulnerability into" in the updated manuscript.

L74: What do you mean by "overall place vulnerability"?

> "Overall place vulnerability" means vulnerability covering the whole study area, it also can be called "the whole vulnerability". Cutter (1996) uses the term "overall place vulnerability" and we added another citation, i.e., Fuchs and Thaler, (2018) that uses the identical term.

L75 Add "critical" before "infrastructure"

> The word "critical" was added before "infrastructure" in the updated manuscript.

L79: Change to "Before 1990s,...was paid...were carried out..."

> We changed to "Before the 1990s,...was paid...were carried out..." on L79. We removed "..before 1990..." due to redundancy.

L81: Change to "However, large losses of life and property resulting from the occurrence of more devastating disasters have brought up the attention on the role of social vulnerability in disaster impact."

> We changed to "However, large losses of life and property resulting from the occurrence of more devastating disasters have brought up the attention on the role of social vulnerability in disaster impact."

L86: Change "management" to "assessment". Management involves not only evaluation.

> We changed "management" to "assessment" in L86.

L86-87: Change to "Hence, governments should analyse...policies such as...to improve its adaptation capacity..."

> We changed to "Hence, governments should analyze...policies such as...to improve its adaptation capacity..."

L88: Change to "considerable amount of research...studies on..."

> We changed to "considerable amount of research...studies on..."

L90-91: Change to "Analysis of SV to storm surges...is important due to four main reasons"

> We changed to "Analysis of social vulnerability to storm surges...is important due to four main reasons" in the updated manuscript.

L91 "Firstly,...few assessments of...in which...is considered"

> We changed to "Firstly,...few assessments of...in which...is considered" in the updated manuscript.

L92: Choose between "detailed" and "comprehensive". They are synonyms.

> We removed "detailed and" and kept "comprehensive" in L92 of the updated manuscript.

L92: What is the object of screening? Could you explain the mechanism of screening?

> First, we consider the continuity and availability of data. Second, we must retain the indicators related to the losses caused by storm surges, such as fishery output value and port cargo throughput. Third, we reserve the indicators that reflect the strength of disaster prevention and mitigation, such as regional GDP. Finally, we retain the weak indicators at the time of the disaster, such as female proportion and total enrollment of students.

L92: The process of screening itself cannot be a buffer against disaster.

> Reply with above paragraph.

L94: Change to "other coastal cities, which are exposed to similar or other types of marine natural hazards." Once more, authors need to specify if and how the methodology was tailored to hazards

resulting from storm surges. As it is, it is applicable to any disaster causing floods (and not only), which means the authors should reconsider the title or give additional emphasis on the hazard intensities and extents related to storm surges; in other words, to further justify their choice of disaster.

> We changed to "other coastal cities, which are exposed to similar or other types of marine natural hazards." in the updated manuscript. Storm surge is the one of the most dangerous natural disasters that happen in Shenzhen. As stated above, we explain the mechanism of screening dedicators. We tested and validated the indicator system using loss data for storm surges and the results are presented in Section **3.3 Validation of SVI to storm surges**.

L95: Change to "Secondly, since 1979, political reform and openness has led...in Shenzhen." Omit "during the study period"

> We made the changes and the sentence now reads "Secondly, since 1979, political reform and openness has led to rapid urbanization and socioeconomic development in Shenzhen."

L95: Omit "expedited process"; "rapid" or "accelerated" is enough to describe the process.

> "expedited process" was removed from L95 and "rapid" was kept to describe the process. The full sentence now reads "Secondly, since 1979, political reform and openness has led to rapid urbanization and socioeconomic development in Shenzhen."

L97: Change to "By choosing Shenzhen, we study a typical scenario of SV change as a result of..."

> We changed to "By choosing Shenzhen, we study a typical scenario of social vulnerability change as a result of..." in the updated manuscript.

L98: "Thirdly, so far,..."

> We changed to "Thirdly, so far,..." in the updated manuscript.

L100: "Instead, herewith, a composite…"

> We changed to "Instead, herewith, a composite…" in the updated manuscript.

L102: "Data envelopment analysis (DEA)...evaluation in China to discover..."

> We changed to "Data envelopment analysis (DEA)...evaluation in China to discover..." in the manuscript.

L105-106: Rephrase "Five methods for combined evaluation were used by Liu and Liu (2017) and results determined that among seven coastal cities selected for evaluation in Shandong Province Yantai city and Binzhou city had the highest and lowest vulnerability, respectively."

> We rephrased to "Five methods for combined evaluation were used by Liu and Liu (2017). Their results determined that among seven coastal cities in Shandong province selected for evaluation, Yantai city and Binzhou city had the highest and lowest vulnerability, respectively."

L108: Rephrase "The socioeconomic vulnerability to typhoon-induced storm surges for municipal districts of Guangdong Province was assessed using the fuzzy comprehensive evaluation. It was determined that vulnerability presented a large spatial heterogeneity (Zhang et al., 2010)."

> We rephrased to "The socioeconomic vulnerability to typhoon-induced storm surges for municipal districts of Guangdong province was assessed using the fuzzy comprehensive evaluation. It was determined that vulnerability presented a large spatial heterogeneity (Zhang et al., 2010)."

L111: Omit "with results from Zhang et al. (2010)"

> We removed "with results from Zhang et al. (2010)" from L111 in the updated manuscript.

L112: Change "differences" to "dimensions".

> We changed from "differences" to "dimensions" in the updated manuscript.

L115: Temporal patterns = trend ?

> We changed from "Temporal patterns" to "trends" in the updated manuscript.

L116: Can you explain what you mean by "macroscopic angle"? Here, you can state again the period your research covers.

> The macroscopic angle means that the scale of the evaluation is based on the whole city but not street or community scales. We changed the sentence to read "...to storm surges in Shenzhen from a macroscopic perspective."

L122: Change to "Since its establishment in 1979, in just 40 years,...through a..."

> We changed to "Since its establishment in 1979, in just 40 years,...through a..." in the updated manuscript.

L124: Change to "However, due to its location at the coast of the Pearl River Delta (Fig. 1a,b) and its proximity to the northern part of the South China Sea (Fig. 1b,c), Shenzhen is facing many coastal disasters threatening its sustainable development, among which storm surge induced disasters are the most severe."

> We changed L124 to read "However, due to its location at the coast of the Pearl River Delta (Fig. 1a,b) and its proximity to the northern part of the South China Sea (Fig. 1b,c), Shenzhen is facing many coastal disasters threatening its sustainable development, among which storm surge induced disasters are the most severe."

L127:"[http://www.sz.gov.cn/ytqzfzx/yingji/yjya/201712/t20171206_10111758.htm (last access: 30 June 2019)]" This should be moved to the list of references.

> This direct URL was already present in the following citation, located in the **References** section, and was removed from L127:

Shenzhen Marine Disaster Emergency Plan, Retrieved from

http://www.sz.gov.cn/ytqzfzx/yingji/yjya/201712/t20171206_10111758.htm (last access: 30 June 2019), 2017

L129: "116 typhoons have seriously affected the Shenzhen coastal area"

> Changed to "116 typhoons have seriously affected the Shenzhen coastal area..." in the manuscript.

L136: Maybe you mean "The increased frequency of storm surges"

> We changed the sentence to read "The increased frequency of storm surges" in the updated manuscript.

L137-138: It is not clear if EWS is a recommendation? Can you relate this to your research results? Which are those "particularly susceptible areas"?

> It is valuable to assess the social vulnerability to storms surges for Shenzhen in order to provide necessary support for the government to improve the level of disaster prevention.

L140: Change "fully contained" to "entirely available"

> We changed from "fully contained" to "entirely available" in the updated manuscript.

L141-142: Change "which was compiled by the Shenzhen Statistical Bureau and a Shenzhen-based investigation team of the National Bureau of Statistics, and published (updated annually) by the Shenzhen Statistical Bureau." to "which is compiled and published on annual basis".

> We changed the sentence from "which was compiled by the Shenzhen Statistical Bureau and a Shenzhen-based investigation team of the National Bureau of Statistics, and published (updated annually) by the Shenzhen Statistical Bureau."
to
"which is compiled and published on annual basis".

L153-154: Change to "Due to the absence of long-term statistical data on some important indicators, this study is limited to a partial statistical dataset spanning the period 1986 - 2016 in order to sustain the data integrity."

> We changed the sentence to "Due to the absence of long-term statistical data on some important indicators, this study is limited to a partial statistical dataset spanning the period 1986–2016 in order to sustain the data integrity."

L165: By "evaluation result", don't you mean "final score"?

> Yes, we meant 'final score'. We changed "evaluation result" to "final score" in the updated manuscript.

L175-178: Provide citations for all mentioned methods and tests.

> Thanks for your suggestion, as citations have been added to the updated manuscript. Specifically, we used Das et al., 2018, Zhou and Yang, 2019, Kuo, 2017, Zhou et al., 2004 and Wen and Hu,

2002 as cited material.

L179&201: Method or strategy?

> Strategy has been deleted. The correct expression is "combination weighting method"

L179-181: Rephrase as follows "Finally, the combined evaluation results are achieved,which have significant advantages compared to those of all single methods due to weighted value of each evaluation method."

> We rephrased the sentence as "Finally, the combined evaluation results are achieved, which have significant advantages compared to those of all single methods due to weighted value of each evaluation method."

L183-184: It is not clear if the first sentence describe a statement or research action.

> The first sentence is a research action.

L187: I cannot understand what the meaning of "knowledge simplicity attribute of rough set" is.

> It means that rough set theory can simplify knowledge and extract the main information from it. The sentence now reads "Finally, the evaluation indicators are screened and the optimal evaluation index system is constructed by using the information extraction ability of rough set."

L188: Among which?

> We deleted "among them" from the beginning of the sentence.

L199: Change "all above evaluation methods" to "all evaluation methods in use"

> We changed from "all above evaluation methods" to "all evaluation methods in use".

L201: Using "a single evaluation framework", do you refer to combined weighting method?

> Yes, we refer to the combination weighting method. The sentence now reads "The results under a single evaluation framework (i.e., the combination weighting method) will be further investigated."

L213: Change to "Calculate the proportion of the indicator j in year i (rij).

> We changed to "Calculate the proportion of the indicator $j$ in year $i$ (rij)." in the updated manuscript.

L234: The meaning of the second sentence in the paragraph is not clear.

> We changed the second sentence and it now reads "When the value of an indicator can clearly distinguish each sample, the indicator possesses resolved information about this evaluation.". Also, we reduced the subsections in this part of the paper (e.g. **2.3.2.3 Coefficient of variation method** becomes **2.2.3 Coefficient of variation method**).

L260: Due to limitations of the methods in use, each single evaluation can lead to a different conclusion."

> We changed from "Due to limitations of the various methods, different single evaluation methods have distinct conclusions."
to
"Due to limitations of the methods in use, each single evaluation can lead to a different conclusion."

L260: Use "Nevertheless" instead of "However". Section 2.4 is really hard to follow. Could you better explain the connection between methods (section 2.3) and resulting indicators (section 2.4)?

> We used "Nevertheless" instead of "However" in the updated manuscript. We have adjusted these sections of the paper.

L286-288: It is not clear if this is achieved within previous researches or is a stage within the present study.

> It is achieved within the present study and we used previous researches as reference.

L289: Storm surges are caused by the action of tropical and extratropical cyclones not accompanied by them.

> This sentence was removed in L289. Yes, you are correct.

L290: Could you explain better the screening process? What is the reason to take out of consideration man-made barriers?

> This is explained above for L92. Urban fixed asset investments may reflect the consideration of man-made barriers.

L291: To what disaster body do you refer: the city itself or a body in general? Which are the bodies you screen?

> We fixed this discrepancy and the sentence now reads "The algorithm screens for classifying a disaster body of interest (i.e., Shenzhen, China) that impact the social economy of the study area and screens for determining key attributes that can affect the exposure of a disaster body."

L291-293: The sentences seem to repeat one another. Could you clarify and rephrase.

> We deleted "As for the exposure of a disaster body, this research selects key indicators that are highly accessible and can reflect a disaster-stricken area at a macro level."

L304: Change "While regional GDP" to "Since the amount of regional GDP"

> We changed from "While regional GDP" to "Since the amount of regional GDP".

L305: Change "equates" to "corresponds", "for" to "to". The last two sentences in section 2.4.1 should change their places.

> We changed "equates" to "corresponds" and "for" to "to". The last two sentences in the third paragraph of Section **2.3 Indicator system of social vulnerability evaluation** (i.e., old **Section 2.4.1**) are now switched.

L312: Again, is Shenzhen city the disaster body? Industries of primary importance for Shenzhen city are…

> This was explained above for L291.

L313: Change "fluctuations" to "changes"

> We changed "fluctuations" to "changes" in the updated manuscript.

L315: Change "higher winds and precipitation patterns" to "severe winds and precipitations"

> We changed "higher winds and precipitation patterns" to "severe winds and precipitations"

L316: Change "inconvenient" to "busy"

> We changed "inconvenient" to "busy" in the manuscript.

L317: Change "suffer casualties outside" to "suffer injuries or even cause casualties"

> We changed from "suffer casualties outside" to "suffer injuries or even cause casualties".

L322: Change "with which" to "meaning that"

> We changed from "with which" to "meaning that".

L324: Change "aspects" to "groups"

> We changed "aspects" to "groups" in the updated manuscript.

L326: Change "more money is devoted into" to "more resources are provided/spent for"

> We changed "more money is devoted into" to "more resources are provided/spent for".

L330: Add "consequences" after "resists disaster"

> We added "consequences" after "resists disaster" in the updated manuscript.

L331: "per capita"

> We changed to "per capita" in the updated manuscript.

L333: Change "infrastructure construction" to "public services"

> We changed from "infrastructure construction" to "public services" in the updated manuscript.

L334-335: Change to "level of medical and health care, including the number of medical and health institutions and their equipment (e.g. beds etc...) as well as the number of health employees."

> We changed to "level of medical and health care, including the number of medical and health institutions and their equipment (e.g. beds, etc.) as well as the number of health employees."

L336: "potential victims"

> We changed the text to "potential victims" in the updated manuscript.

L342: Change "social vulnerability to storm surges discussed in this research can be approximately divided into" to "degrees of social vulnerability to storm surges discussed in this research are set to..."

> We changed "social vulnerability to storm surges discussed in this research can be approximately divided into" to "degrees of social vulnerability to storm surges discussed in this research are set to..."

L345: It is not clear if proposed SVI threshold values were calculated by the authors or were borrowed from Yuan et al. (2016).

> We changed the standard and calculated the threshold values by ourselves.

L346: Change "close" to "similar"

> We changed "close" to "similar" in the updated manuscript.

L422: Change "obvious" to "pronounced", "variation" – "variability"

> We changed "obvious" to "pronounced" and "variation" to "variability" in the manuscript.

Figure 2 is not referred to in the text.

> We referred to Figure 2 in the last sentence of the first paragraph of **Section 2.1**.

L753: Change to "and outlined using crimson colour."

> We changed to "and outlined using crimson color." in the updated manuscript.

L836: Change to "method (blue line).  The weighted value of SVI is depicted with thick red line."

> We changed to "method (blue line). The weighted value of SVI is depicted with thick red line." in the updated manuscript.

Table1/3: Resilience: Per capita

> We changed from "Resilience" to "Resilience: Per capita" in both Table 1 and Table 3.

**Confirmed additions:**

Added a new paragraph #2 to Section **1. Introduction**.
> This paragraph discusses the issue of storm surges in China from an economic and spatial perspective.

Added geographic coordinates of Shenzhen, China in Section **2.1 Study area and data sources**.
> The coordinates of (22° 32' 34.3788" N, 114° 3' 46.7856" E) were added using DMS notation.

Sections **2.1 Study area** and **2.2 Data sources** have been merged.
> There is a new section **2.1 Study area and data sources**

The words "Province" in the manuscript are now lower case.
> The word is now shown as "province" when naming specific provinces.

Subsections in **2.x** were reduced to eliminate tertiary subsections (e.g. **2.3.2.2** TOPSIS method was changed to **2.2.2** TOPSIS method).
> This is complete.

Subsection reduction to **3.1 Variation characteristics of social vulnerability**
> We removed subsections **3.1.1 Interannual variation** and **3.1.2 Interdecadal variation** and combined those paragraphs to section **3.1 Variation characteristics of social vulnerability**.

Subsection reduction to **3.2 Reasons for vulnerability changes**
> We removed subsections **3.2.1 Analysis of resilience changes**, **3.2.2 Analysis of exposure and sensitivity changes** and **3.2.3 Correlation between value of indicators and SVI**. All of those paragraphs are under **3.2 Reasons for vulnerability changes**.

Added parts to sentences
> Added parts "(–0.006 per year)" and "(–0.04 per year)" to the following phrases "...SVI illustrates a significant downward trend (–0.006 per year) in entirely..." and "...shows a significant downward trend (–0.04 per year) for the remaining years...".

Added parts to sentence
> Added part "...in which better protected buildings and factories have been built in what used to be farmland,..." to phrase "...high-speed development for a second moment, in which better protected buildings and factories have been built in what used to be farmland, causing the proportion of agriculture..."

Added word to sentence
> Added "Consequently,..." to phrase "Consequently, the total sown area..."

New section added to manuscript named **3.3 Validation of SVI to storm surges**.
> This section adds a validation study and 2 new figures (i.e. Figure 8 and 9) to the research.

Added equation (11) to section **3.3 Validation of SVI to storm surges**
> It is cited several times in the neighboring paragraph.

Added (i), (ii) and (iii) to third paragraph to Section **4. Conclusion**.

> This is complete.

Added new sentences and sections to existing sentences in section **4. Conclusion**
> Added part "...validated to be…" to sentence "The final weighted SVI is validated to be rational and reliable by combining results from multiple evaluation methods, based on the idea of combination weighting, in order for the results to objectively reflect the connotative information of social vulnerability in the indicator system."

> Added sentence "This paper successfully evaluates the social vulnerability to storm surges from a macroscale perspective using 30 years of economic statistical data and 24 years of loss data."

> Added part "coastal breakwaters, flooding areas," to sentence "However, some indicators were not included in the final evaluation system due to the lack of statistical data, such as coastal breakwaters, flooding areas, insurance depth and housing values."

> Added part "it is obvious that...", added part "is not as granular..." and removed part "cannot be substituted for the vulnerability differences at" from the sentence "Additionally, it is obvious that the scale of the social vulnerability evaluation at the municipal level is not as granular as administrative units smaller than the municipal level, such as districts, towns and streets."

Data Availability: Modification and updated
> The paragraph has been updated to account for the new Figures 8 and 9.
> Updated this section to account for the loss data from the Ministry of Natural Resources in the Bulletin of China Marine Disaster, including URLs.

Author contributions: Modification
> Modified the **Author contributions** section to reflect the changes in work ownership, new figures and figure descriptions.

Acknowledgements: Updated
> Updated this section with a new grant number and removed the following part ", the Fundamental Research Funds for the Central Universities (Grant Nos. 3001000-841564014, 3006000-841762015, 201562030)".

References: Added data reference
> Added the following dataset reference as well as cited it in the manuscript:
Bulletin of China Marine Disaster, Ministry of Natural Resources of the People's Republic of China, Retrieved from
http://www.mnr.gov.cn/sj/sjfw/hy/gbgg/zghyzhgb/ (last access: 08 April 2020), 2018.

Figure 2: Replacement
> Replaced original Figure 2 with a Python version Figure 2, for consistency among several other plots.

Figures 8 and 9: Added
> In support of the validation section, two figures were created and added to the manuscript including appropriate captions.  These figures were generated with MATLAB scripts.

General formatting to clean up the final PDF version.
> This is complete.

[revised manuscript text omitted]

---

## Referee Report (RR1)

Comments on "Trends in social vulnerability to storm surges in Shenzhen, China" by Yu et al.

The authors mentioned few limitations of this study in their response. It might be beneficial to include these limitations in the manuscript. For example, the authors' response to "Page 11, lines 314-319" (Now Pages 11-12, lines 322-326).

Figures 6, 7: The authors could include the normalization approach that they used in the manuscript, as there are several normalization approaches.

---

## Author Response (AR2)

**Author's Response to Anonymous Reviewer #1**

The research group thanks Anonymous Reviewer #1 again for their additional suggestions. The feedback allows our group to make important updates and to build an improved version of the manuscript.

An author's response to Anonymous Reviewer #1 is denoted with a > symbol and blue text. Some confirmed additions are placed in magenta text and are located at the end of the Author's Response to Anonymous Reviewer #2.

**Reviewer's suggestions:**

The authors mentioned few limitations of this study in their response. It might be beneficial to include these limitations in the manuscript. For example, the authors' response to "Page 11, lines 314-319" (Now Pages 11-12, lines 322-326).

> We added the following sentences to the final paragraph under section **2.1 Study area and data sources** (i.e. L166).
"Although including all factors to the indicator system for analysis would reach better agreement with the marine disaster community, this study can only provide certain factors due to data availability limits. For example, elderly people and people with disabilities are included in vulnerable groups (Yuan et al., 2016) which should be reflected in sensitivity, but there is no specific data captured about the elderly population in Shenzhen's statistical yearbooks."

> We added the following sentence to the final paragraph under section **2.1 Study area and data sources** (i.e. L166).
"In terms of study areas, the research limits coastal city choices based on several assumptions. Candidate cities should have (i) datasets with relatively complete, detailed statistics, (ii) well developed coastal industries such as agriculture, fishing, etc., (iii) a sharp, increasing population growth and matching economic development pattern, and (iv) suffer from frequent and severe storm surges."

> We added the following sentences to the final paragraph under section **2.1 Study area and data sources** (i.e. L166).
"Additionally, non-candidate coastal cities are mature, populous cities with a long economic history and had slower development stages or primitive cities with a slower economic growth rate and possess fewer established coastal industries. As a limit to the study, a fit method should be developed to determine which cities match specific criteria suitable for becoming appropriate candidates for this research."

> We added the following sentence to the section **3.3 Validation of SVI to storm surges** (i.e. L414).
"Economic loss data due to storm surges in Shenzhen is unavailable and a broader scale dataset was used in the validation of SVI to storm surges."

> We added the following sentence towards the end of section **4 Conclusion** (i.e. L473), which relates to the sentences at L209-211 in section **2.2 Research methods**.
"To further increase the reliability of the social vulnerability evaluation results, additional methods (e.g., fuzzy cluster analysis, PCA, efficacy coefficient method, expert evaluation method, etc.) and a greater number of methods deployed should be included in the research."

Figures 6, 7: The authors could include the normalization approach that they used in the manuscript, as there are several normalization approaches.

> For **Fig. 6**, we changed the caption to read "Normalized values of total area of crops (yellow line) and proportion of females (green line). Note, the min-max normalization method carries out a linear transformation on the original dataset to standardize each row into an interval $[y_{min}, y_{max}]$ using the formula: $y = (y_{max} - y_{min})*(x - x_{min}) / (x_{max} - x_{min}) + y_{min}$. These results fall in the interval [0, 1]."

> For **Fig. 7**, we changed the caption to read "Three most relevant indicators of social vulnerability during the research period. SVI is shown in red dots. The min-max normalization method used in **Fig. 6** was used in this figure and the results fall in an interval [0, 0.25]. SVI values were subtracted by a constant (0.38) to meet an identical interval. Note, the y-axis is partially visible to expand the lower portion of the plot."

**Author's Response to Anonymous Reviewer #2**

The research group thanks Anonymous Reviewer #2 again for their detailed review, valuable comments and constructive feedback. The group found all of the feedback helpful in creating a higher quality manuscript.

An author's response to Anonymous Reviewer #2 is denoted with a > symbol and blue text. Some confirmed additions are placed in magenta text and are located at the end of this response.

**General comments:**

As stated in the first review, the proposed research presents a cogent effort to assess the long-term trend in the social vulnerability to storm surge induced flood hazard in Shenzhen, China using a system of SV indicators. Particularly valuable is the complex approach used for the SVI construction, which is a combination and weighting of results obtained through application of three single evaluation methods. It is reconfirmed that the work is interesting in the context of preparedness, mitigation and adaptation of a large city to natural disaster impacts. Such a study is certainly useful for wide range of stakeholders, especially policy makers, local authorities and coastal managers.
The group is pleased to hear that the research will be useful to a wide range of scientific professionals.

The authors have addressed the points raised in the first review in a satisfactory way by:
1) adding more detailed information on storm surge-induced flood hazard intensity and extent and about the most susceptible coastal areas;
2) substantiating the geographical scope of the study revealing that "most areas of Shenzhen are threatened during storm surges". I consider their argument "This work is a creative attempt to analyze publicly available "macroscopic" data in order to explain the "microscopic" phenomena for such similar Chinese coastal cities" as valid. The work therefore is of considerable scientific and practical importance;
3) providing relevant citations.
4) The Results & Discussion chapter appear to be more balanced with respect the previous one;
5) The text is much more fluent and easy to follow; the use of English was considerably improved.

A general remark about the figures' captions: some of them are quite short and not explanatory enough. Therefore, it is advisable to extend the captions' content as much as possible.

> We changed the caption of **Fig. 3** to read "Basic four-step procedure (colored boxes) in calculating SVI (black box). The second step (rose boxes) uses three separate methods, while the third (orange box) and fourth (green box) steps are meant to integrate the three calculated results of the second step. Note, the black dashed box surrounding SVI indicates a result of the four-step process."

> We changed the caption of **Fig. 6** to read "Normalized values of total area of crops (yellow line) and proportion of females (green line). Note, the min-max normalization method carries out a linear transformation on the original dataset to standardize each row into an interval $[y_{min}, y_{max}]$ using the formula: $y = (y_{max} - y_{min})*(x - x_{min}) / (x_{max} - x_{min}) + y_{min}$. These results fall in the interval [0, 1]."

> We changed the caption of **Fig. 8** to read "Standardized SVI (green line), intensity (blue line) and loss (red line) from 1991 to 2015. Note, (i) the use of min-max normalization and (ii) the range for intensity and loss is unified to the interval [0.4, 0.7] for a convenient comparison with SVI."

**Technical corrections:**

L20: "having the most considerable economic development potential"

> We changed to "having the most considerable economic development potential"

L21: Use "for evaluation of its"…. "spanning the period 1986 – 2016".

> We changed to "for evaluation of its" … "spanning the period 1986–2016".

L22: use "leading to some alterations (or variations) of the social vulnerability trend"

> We changed to "leading to some alterations of the social vulnerability trend"

L24: use "afterwards" instead of "in the rest of the time"

> We replaced "in the rest of the time" with "afterwards"

L30: use "to an abnormal…"

> We used "to an abnormal..."

L36: use "related" or "resulting" instead of "their"

> We used "resulting" and the phrase now reads "by resulting storm surge, ..."

L39: "losses and casualties resulting from"

> We used "losses and casualties resulting from…."

L40: "governments and local authorities"… "disaster risk prevention…"

> We used the phrases "governments and local authorities" … "disaster risk prevention…"

L43: use "Both types of …"

> We used "Both types of ..."

L44: Maybe you mean "susceptible to" by using "suitable"

> Changed from "is very suitable for the" to "is very susceptible to the development..."

L46: "often occur"

> Changed from "frequently occur" to "often occur"

L47: use "suffers" instead of "receives"

> Changed from "receives" to "suffers"

L50-51: "The spatial distribution of storm surge disasters shows that Guangdong, Zhejiang, Fujian and Hainan are the most affected provinces."

> Changed sentence to "The spatial distribution of storm surge disasters shows that Guangdong, Zhejiang, Fujian and Hainan are the most affected provinces."

L53: ", respectively, …"

> Changed from ", respectively ..." to ", respectively, …"

L59: "Risk assessment of tropical.."

> Changed to "Risk assessment of tropical…."

L60-61: "An effective coping with disaster risk requires a more rational distribution of efforts in areas such as disaster risk reduction and disaster management."

> Changed sentence to read "An effective coping with disaster risk requires a more rational distribution of efforts in areas such as disaster risk reduction and disaster management."

L63: use "real" or "tangible" instead of "realistic"

> Changed "realistic" to "tangible"

L67: "Therefore, the ability…"

> Changed to "Therefore, the ability…."

L91: "It is…"

> Changed to "It is…."

L104: rephrase "last decade" as it is used for publication from 2003-2008.

> We rephrased to "same period"

L105: It is a bit unusual to refer to Figs in the Introduction. You may omit Fig. 1c and first mention it in the study site description, e.g. L135-138, in any case before Fig.2

> Removed "(Fig. 1c)" from L105 and added "(Fig. 1c)" to the end of the second sentence in the first paragraph under **2.1 Study area and data sources**

L113: instead of "capable" you can use "developed" or similar

> We changed from "capable" to "developed"

L126: "for one disaster prone coastal city..."

> Changed to "for one disaster prone coastal city….."

L139: "during the study period"

> We changed to "during the study period"

L142: "severe."

> We fixed the error from "severe.." to "severe."

L145: "within the city limits"

> We changed the phrase to "within the city limits"

L147: "Typhoon "7908" made landfall in the end of July 1979, which caused the storm surge level at Red Harbor…"

> We changed to "Typhoon "7908" made landfall in the end of July 1979, which caused the storm surge level at Red Harbor…."

L151: "caused ever growing…"

> We changed the phrase to "caused ever growing…."

L153: "future storm surge impacts"

> We changed to "future storm surge impacts"

L186: "above mentioned"

> We changed from "above predecessors'" to "above mentioned"

L216: "the smaller the uncertainty and the entropy"

> We changed to "the smaller the uncertainty and the entropy"

L221: "according to their variation degree, using information.."

> We changed to "according to their variation degree, using information…."

L243: "The TOPSIS method is performed in six steps"

> We changed from "can be divided into" to "is performed in", and the phrase now reads "The TOPSIS method is performed in six steps"

L305: "aspects of both population and industrial structure"

> We changed to "aspects of both population and industrial structure"

L306: "degree of sensitivity"

> We changed to "degree of sensitivity"

L308-309: "Grade I indicators are identified with the three components of vulnerability and the Grade II indicators – with the branches of the Grade I indicators."

> We changed to "Grade I indicators are identified with the three components of vulnerability and the Grade II indicators – with the branches of the Grade I indicators."

L320: "will directly affect their output."

> We changed to "will directly affect their output."

L323-324: "Representing vulnerable societal groups, students and women are more likely to be injured or even to suffer casualties"

> We changed to "Representing vulnerable societal groups, students and women are more likely to be injured or even to suffer casualties..."

L330: "reflect the general public"

> We changed to "reflect the general public"

L331: "infrastructure development."

> We changed to "infrastructure development."

L333: "the more developed the regional infrastructure is."

> We changed to "the more developed the regional infrastructure is."

L344: You might consider using "variation pattern"

> We changed the subsection title to "**3.1 Variation pattern of social vulnerability**"

L350: "corresponding critical thresholds"

> We changed to "corresponding critical thresholds"

L355: "As shown in Fig. 4, the weighted SVI exhibits a well pronounced overall downward trend (–0.006 per year) with noticeable fluctuations"

> We changed the phrase to read "As shown in Fig. 4, the weighted SVI exhibits a well pronounced overall downward trend (–0.006 per year) with noticeable fluctuations"

L361: "Thus, the high social vulnerability stretched over the longest period of time opposed to the low vulnerability, which was only observed during the last two years of the study period."

> We changed the sentence to read "Thus, the high social vulnerability stretched over the longest period of time opposed to the low vulnerability, which was only observed during the last two years of the study period."

L364: "each decade represents a cycle"

> We changed to "each decade represents a cycle"

L366: "The discovered trend…" … "recover after substantial damage"

> We changed the phrases to read "The discovered trend…." … "recover after substantial damage"

L372: "corresponding indices"

> We changed to "corresponding indices"

L380: "the continuous increase of resilience is the most significant feature…"

> We changed the phrase to "the continuous increase of resilience is the most significant feature…"

L384-386: "EI remains almost constant during the period 1986 -1991 and, after presenting a slight drop between 1992 and 1996, continues growing." … " Shenzhen transformed from a small fishing village to grids of high-rise buildings after the rapid urbanization that followed the reform and openness policy occurred in 1979. This has led to a continuous decreasing trend of the exposure indicator (Fig. 6)."

> We changed the sentences to read "EI remains almost constant during the period 1986–1991 and, after presenting a slight drop between 1992 and 1996, continues growing since 1996." … " Shenzhen transformed from a small fishing village to grids of high-rise buildings after the rapid urbanization that followed the reform and openness policy occurred in 1979. This has led to a continuous decreasing trend of the exposure indicator (i.e., total sown area of crops; Fig. 6)."

L389-390: "Shenzhen entered the second stage of speeded economic growth, during which better protected buildings and factories were built on what used to be farmland."

> We changed the sentence to "Shenzhen entered the second stage of speeded economic growth, during which better protected buildings and factories were built on what used to be farmland, ..."

L392: Omit "simultaneously" … "the weight of the 'total sown area of crops' indicator was relatively large"

> We removed "simultaneously" and we changed the phrase to read "the weight of the 'total sown area of crops' indicator was relatively large"

L394: use "lowest" instead of "slowest".

> We changed "slowest" to "lowest"

L394: The phrase "SI maintains an upward trend until 2000 to 2011 when the trend exhibits an oblate form" is not clear.

> We changed the sentence to "..., SI maintains an upward trend except for a small decline between 2001 and 2006..."

L396: Rephrase, e.g. "showed a significant decreasing trend until 2006, which than sharply increased in a 10-year period"

> We changed "...showed a significant decreasing trend firstly and then increased..." to "...showed a significant decreasing trend until 2006, which than sharply increased in a 10-year period"

Fig. 6: female 'scale' or 'proportion'; it's better using 'proportion of female population' or 'proportion of females" instead.

> The caption for Fig. 6 was changed from "female proportion" to "proportion of females".  In Fig. 6, the dark green line was renamed "Proportion of Females" instead of "Female Scale".  The figure was reproduced.

L398: "weight of the indicators by benefit and cost types"

>  We changed to "weight of the indicators by benefit and cost types"

L398: Do you mean "similar" by using "proximate" or "closest in relationship; immediate (especially of the cause of something)"?

> We changed this part of the sentence to read "is very similar,….".  We meant the indicators were very close in numerical value.

L400: "The statistical data corresponding to the resilience indicators are generally larger…"

> We changed the phrase to read "The statistical data corresponding to the resilience indicators are generally larger…"

L402-403: Rephrase as "is a measure of degree of influence of this indicator on the social vulnerability".

> We rephrased to "while the correlation coefficient between the indicator value and SVI is a measure of degree of influence of this indicator on the social vulnerability."

L403: "are determined to be the number..."

> We changed to "are determined to be the number..."

L407: "act a secondary"

> We changed to "act a secondary"

L420: "The best fit equation reads:"

> We changed to "The best fit equation reads:"

L423: omit "at heightened levels"

> We removed the part "at heightened levels"

L427: "reliability of Eq. (11) is considered high"

> We changed to "reliability of Eq. (11) is considered high"

L428: omit "from the analysis" or rephrase

> We decided to remove "From the analysis,"

Conclusion: Move the last sentence of the first paragraph as follows: "This research evaluates social vulnerability to storm surges in Shenzhen, China, from a macroscale perspective using 30 years of economic statistical data and 24 years of loss data."

> We moved the last sentence of the first paragraph and placed it as the first sentence of the first paragraph.  We also corrected the length of years for each dataset.  The sentence now reads "This research evaluates social vulnerability to storm surges in Shenzhen, China, from a macroscale perspective using 31 years of economic statistical data and 25 years of loss data."

L432: omit "Then," in the second sentence.

> We removed "Then,"

L440: "The trend experiences four stages, passing through high to low social vulnerability"

> We changed to "The trend experiences four stages, passing through high to low social vulnerability"

L443-444:"from exposure, sensitivity and resilience perspective, it is revealed that the increase of the social economy exposure and demographic and industrial structures sensitivity are less important than the disaster resilience"

> We changed the phrase to read "from exposure, sensitivity and resilience perspective, it is revealed that the increase of the social economy exposure and demographic and industrial structures sensitivity are less important than the disaster resilience"

L445: "while the capacity to withstand and response to disasters has significantly improved"

> We changed to "while the capacity to withstand and response to disasters has significantly improved"

L448: "The study concludes that the increase of residents' income, infrastructure enhancement and medical and health conditions improvement …"

> We changed to "The study concludes that the increase of residents' income, infrastructure enhancement and medical and health conditions improvement …"

L459: "reasonable arrangements"

> We changed to "reasonable arrangements"

L461: "but their growth rate"

> We changed to "but their growth rate"

L468: "departments should assess all aspects of the damage"

> We changed to "departments should assess all aspects of the damage"

L470: "disaster risk prevention, preparedness and reduction"

> We changed the phrase to read "disaster risk prevention, preparedness and reduction"

L473: Use "as detail as smaller administrative units, such as districts, towns and streets". I am not sure if town is smaller disaggregation than district. You may use "residential quarter" instead.

> We changed the phrase to read "as detailed as smaller administrative units, such as districts, residential quarters and streets"

L475: Rephrase as: "Further challenges are related to narrowing of the evaluation scale of social vulnerability and selection of more reasonable indicators according to the local conditions."

> We rephrased to the following "…, further challenges are related to narrowing of the evaluation scale of social vulnerability and selection of more reasonable indicators according to the local conditions."

**Additional Changes**

L14: Addition

> We added an Orcid account as "HY$_7$, https://orcid.org/0000-0002-0529-1172".  The subscript near the initials is the position of this author in the author list.

L24: Revision

> We changed from "the past 30 years." to "the past 31 years." because we count each year as a whole.

L341: Revision

> We changed from "(e.g. beds, etc.)" to "(e.g., beds, etc.)"

L395: Revision

> We rephrased from "the indicator of female proportion…." to "the 'proportion of females' indicator"

L395-396: Revision

> We rephrased from "the indicator of female proportion..." to "the proportion of females indicator..."

L415: Removed phrase

> We removed "Due to the lack of data in Shenzhen, ..."

L432: Removed sentence.

> We removed the second sentence "This research evaluates social vulnerability to storm surges in Shenzhen, China." because it is redundant with the new first sentence of the same paragraph.

L438: Revision

> We changed from "historical 30-year dataset…." to "historical 31-year dataset…."
> We changed from "24 years of loss data." to "25 years of loss data." because we count each year as a whole.

Data availability: Revision

> We changed from "30-year dataset…." to "31-year dataset…." on L480
> We changed from "24-year dataset…." to "25-year dataset…." on L484

Figure 3: Replacement

> We changed the box colors to show the 4-step process in calculating SVI (i.e. the result).  We used a neutral color for SVI (black) with a black dashed line surrounding it to indicate that SVI is the result (not a step).  The new figure was regenerated and replaced the original **Fig. 3**.

Figure 6: Replacement

> Fig. 6 was reproduced.  The dark green line was renamed to "Proportion of Females" instead of "Female Scale".

[revised manuscript text omitted]

---

## Author Response (AR3)

**Additional Changes**

A few revisions were made to the accepted manuscript and are outlined below. The changes are denoted with a > symbol and blue text. The corrected manuscript is appended below showing those changes.

*The line numbering is based on manuscript file: nhess-2019-293-manuscript-version3.pdf

L2-3: Revisions (Author list)

> Removed two coauthors from the Author list.
> Changed several superscripts to the Author list for correct affiliations.
> Remove the superscript [†] from Huaming Yu and Yuhang Shen.
> Rearranged the new Author list as the following "Huaming Yu[1,2,3], Yuhang Shen[1], Ryan M. Kelly[4], Haiqing Yu[5], Xin Qi[6], Songlin Li[1]"

L5-9: Revisions (Author affiliations)

> Added "[2]Sanya Oceanographic Institute, Ocean University of China, Sanya, 572024, China;" to the affiliations list (L5).
> Shifting the superscript numbering [2-5] due to affiliation addition.
> Changed "Key Laboratory of Physical Oceanography, Ministry of Education, Qingdao, 266003, China;" from superscript [2] to superscript [3]
> Changed "Rykell Scientific Editorial, Los Angeles, CA, USA;" from superscript [3] to superscript [4]
> Changed "Department of Organic Food Quality and Food Culture, Faculty of Organic Agricultural Sciences, University of Kassel, Nordbahnhofstrasse 1A, 37213, Witzenhausen, Germany;" from superscript [4] to superscript [6] and moved the affiliation to the correct position.

L11: Remove statement

> Removed the statement "† These authors contributed equally to this work."

L12: Revision (Corresponding author)

> Change corresponding author from "*Correspondence to*: Huaming Yu (hmyu@ouc.edu.cn)" to "*Correspondence to*: Haiqing Yu (yuhaiqing@ouc.edu.cn)"

L15: Revision (Author subscript)

> Changed Haiqing Yu's subscript from [7] to [4], due to the author position change (L15).
> Moved "HY[4], https://orcid.org/0000-0002-0529-1172" from L15 to L14 for ordering purposes.

L504-513: Revisions (Author contributions)

> Removed the initials KW and XB.
> Removed initials KW from L510.
> Changed Haiqing Yu's subscript from [7] to [4], due to the author position change (L510).
> Removed the sentence "XB offered technical guidance and screening of the paper." in L510-511.

L521-524: Revisions (Acknowledgements)

[revised manuscript text omitted]